# Morphological characterization of antennae and antennal sensilla of *Diaphorina citri* Kuwayama (Hemiptera: Liviidae) nymphs

**Lixia Zheng[1,2], Qichun Liang[1], Ming Yu[1], Yi Cao[1], Wensheng Chen[1]***

**1** Department of Horticulture, Foshan University, Foshan, China, **2** College of Agronomy, Jiangxi Agricultural University, Nanchang, China

* cjf0000@163.com

**Data Availability Statement:** All relevant data are within the manuscript and its Supporting Information files.

## Abstract

*Diaphorina citri* Kuwayama is the most economically important citrus pest which is the primary vector of *Candidatus* Liberibacter spp. causing citrus greening (huanglongbing, HLB) disease. To better understand the developmental and structural changes of antennae and antennal sensilla in *D. citri* nymphs, we investigated the antennal morphology, structure and sensilla distribution of the five nymphal stages of *D. citri* using scanning electron microscopy. The antennae of the five different nymphal stages of *D. citri* were filiform in shape, which consisted of two segments in the first-, second- and third-instar nymphs; three segments in the fourth- and fifth-instar nymphs. The length of their antennae was significantly increased with the increase of the nymphal instar, as well as the total number of antennal sensilla. Ten morphological sensilla types were recorded altogether. They were the long terminal hair (TH1), short terminal hair (TH2), sensilla trichoidea (ST), cavity sensillum 1 (CvS1), cavity sensillum 2 (CvS2), sensilla basiconica 1–3 (SB1-3), sensilla campaniform (SCA) and partitioned sensory organ (PSO). Also, the distribution of antennal sensilla in each nymphal stage of *D. citri* was asymmetrical. The SBs only occurred on the antennae of the third-, fourth- and fifth-instar nymphs. Only one CvS2 was found in the third- and fifth-instar nymphs, and one SCA in the fourth- and fifth-instar nymphs, respectively. The possible roles of the nymphal antennal sensilla in *D. citri* were discussed. The results could contribute to a better understanding of the development of the sensory system, and facilitate future studies on the antennal functions in *D. citri* nymphs.

## Introduction

The Asian citrus psyllid (ACP), *Diaphorina citri* Kuwayama (Hemiptera: Liviidae), is the most economically important citrus pest because it is the primary vector of *Candidatus* Liberibacter spp. causing citrus greening (huanglongbing, HLB) disease, also the most serious citrus disease worldwide [1,2]. *Diaphorina citri* apparently originated in southern Asia, but now has spread to other world citrus-producing areas [2]. More than 50 species of the family Rutaceae have been recorded as hosts of *D. citri* [3]. Nymphs and adults suck phloem sap from the foliage

**Funding:** This research was supported by the National Natural Science Foundation of China (31660517) and the Natural Science Foundation of Jiangxi Province, China (20161BAB214172).

**Competing interests:** The authors have declared that no competing interests exist.

causing leaf distortion, curling and deposition of honeydew on leaves that result in mold growth [3,4]. However, the most serious threat to citrus worldwide comes from its ability to transmit the bacteria, *C. Liberibacter* spp., associated with HLB which causes yellowing of shoots, stem dieback, sour fruit, crop losses and eventually tree death [5,6].

For most insects, the antennal sensilla are peripheral sensory structures involved in habitat searching, host location, host discrimination, mating and oviposition [7–9]. They carry different kinds of sensilla which can be classified as chemosensory, mechanosensory and thermo-hygroreceptive sensilla based on their morphological and functional characteristics [7,10]. Studies on the interaction between ACP and host plant volatiles indicated the critical role of its olfactory system in finding host plants [11,12], mating [12–14], and oviposition [13,15]. Coutinho-Abreu et al. have recognized the ACP antennal neurons that responded strongly to odorants found in the host citrus plants using single-unit electrophysiology [16]. Recently, Zanardi et al. have identified the putative ACP sex pheromone [17], in addition to other related compounds that also generated electroantennographic (EAG) responses [17–19]. Thus, to analyze the antennal morphology, structure and sensilla distribution in *D. citri* is important to explore their life cycle, olfactory behavior and host identification mechanisms.

Previous studies have described the external morphology and ultrastructure of antennal sensilla in adult *D. citri* [20], antennae sensory arrays in adult *D. citri* [21], and antennal transcriptome in adult *D. citri* [22]. However, there is very little information about the morphology of nymphal antennae and antennal sensilla of psyllids. In this study, we investigated the antennal morphology, structure and sensilla distribution in the five different nymphal stages of *D. citri* using scanning electron microscopy. The results could contribute to a better understanding of the development of the sensory system, and facilitate future studies on the antennal functions in *D. citri* nymphs.

## Materials and methods

### Insects

Specimens used in the study were the five different nymphal stages of *D. citri*. These insects originated from a colony maintained at Jiangxi Agriculture University at Nanchang, China (28˚45'36"N, 115˚49'43"E) that was initiated from insects collected in South China Agriculture University at Guangzhou, China (23˚10'7"N, 113˚21'27"E) in June 2016. The ACP were fed on *Murraya exotica* seedlings at 27 ± 1 ºC, 70 ± 5% and 14:10 L:D photoperiod.

### Scanning electron microscopy (SEM)

The whole bodies of the five different nymphal stages of *D. citri* were fixed in 2.5% glutaraldehyde at 4˚C for 24 h and post-fixed in 1% osmium tetraoxide for 2.5 h. The treatment process for the specimens including rinsing, dehydrating and drying was according to Yang et al. [23]. Finally, the specimens were anchored on a holder using double-sided adhesive tape in the ventral and dorsal positions, sputter-coated with gold, examined and photographed either at 10 kV using an SEM (XL30, FEI, Holland and Nova Nano 430, FEI, Holland) or 25 kV by an SEM (JSM-6360LV, Japan).

### Statistical analysis

Classification of sensilla types in this work was based on the morphological characteristics of similar structures described previously [24–32].

Statistical tests were performed with SPSS 17.0. Sensilla on the dorsal, ventral and two lateral surfaces of the five different nymphal stages of *D. citri* antennae were counted and

measured. The means (*n* = 10) were analyzed by general linear model (GLM) procedure and Tukey's mean separation test. Mann-Whitney *U* test was used to compare the length and width of various nymphal stages of *D. citri* between the long terminal hair (TH1) and short terminal hair (TH2).

## Results

### General morphology of antennae

The antennae of *D. citri* nymphs were observed when individuals were fixed in a ventral position and looking at the region located between the compound eyes. The morphology of the five nymphal stages of *D. citri* antennae is illustrated in Fig 1. The antennae of the first-, second- and third-instar nymphal stages of *D. citri* were composed of two segments (Fig 1A, 1B and 1C), whereas the fourth- and fifth-instar nymphal antennae consisted of three parts: two basal segments and the flagellum with poorly defined subsegments (Fig 1D and 1E). There were significant differences found in the length of the antennae of each nymphal stage of *D. citri* (Table 1). The length of the five different nymphal antennae was significantly increased with the increase of the nymphal instar. We also observed significant difference in the growth of the near two nymphal instars (Table 1). Additionally, the fifth-instar nymphal antennae were five times longer than the first-instar nymphal antennae.

### Types of sensilla

Table 2 shows the abundance and distribution of the antennal sensilla in the five nymphal stages of *D. citri*. There were ten morphological sensilla types recorded altogether. They were the TH1, TH2, sensilla trichoidea (ST), cavity sensillum 1 (CvS1), cavity sensillum 2 (CvS2), sensilla basiconica 1–3 (SB1-3), sensilla campaniform (SCA) and partitioned sensory organ (PSO) (Table 2). Moreover, we observed the asymmetrical distribution of the antennal sensilla in the five nymphal instars. The ST, TH1, TH2, PSO and CvS1 were found on the antennae of each nymphal stage of *D. citri*; The CvS2 and SB3 only occurred on the antennae of the third- and fifth-instar nymphs, and one SCA was found in the fourth- and fifth-instar nymphs (Table 2). In addition, the SB1 and SB2 were observed on the antennae of the third-, fourth- and fifth-instar nymphs. We also found the total number of the antennal sensilla increased from the first- to the fifth-instar nymphs.

### Morphology and structure of sensilla

**Terminal hairs.** There were two bristle-like sensilla apically on the terminal antenna of the five nymphal stages of *D. citri* (Fig 1), with an emphasis in the third-instar nymphs. The TH1 was strong and straight with longitudinal grooves (Fig 2A). They were inserted into a big cuticular socket. The TH2 had a similar morphology to the TH1, but much shorter and stronger (Fig 2B, Table 3). The length of TH1 was significantly longer than that of the TH2 in each development stage of *D. citri*. Moreover, the length of TH1 in the second-instar nymphs was significantly shorter than that in the fifth-instar nymphs, but no difference in the other nymphal stages. The length of TH2 in the fifth-instar nymphs was much longer than those in the second- and third-instar nymphs. No difference was found in the width between TH1 and TH2 except those in the fifth-instar nymphs. In addition, the width of TH2 in the fifth-instar nymphs was significantly bigger than that in the other nymphal stages of *D. citri* and that of the TH1 (Table 3).

**Sensilla trichoidea.** Sensilla trichoidea (ST) was found on the antennae of each nymphal stage of *D. citri* (Table 2). They were straight hairs with a smooth surface and a big conical

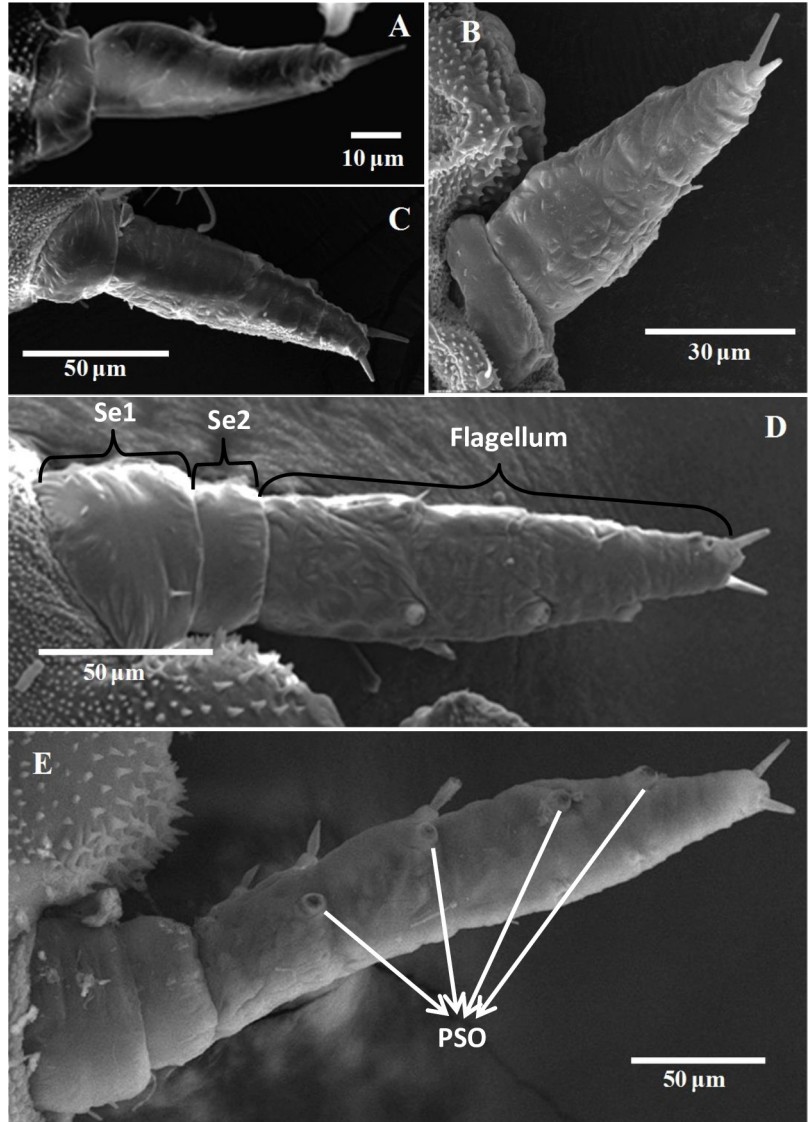

**Fig 1. General morphology of the antennae of the immature *Diaphorina citri*.** (A-C) SEM photographs of the first-, second- and third-instar of *D. citri* nymphal antennae with two segments. (D-E) SEM photographs of the fourth- and fifth-instar of *D. citri* nymphal antennae with three segments. PSO, partitioned sensory organ; Se1, the first segment of the antennae; Se2, the second segment of the antennae.

**Table 1. Length and growth (mean ± SE) of the five nymphal stages of *Diaphorina citri* antennae.**

| Nymphal stages | N | Length (µm) | Growth (µm) |
|---|---|---|---|
| First-instar | 10 | 61.85 ± 1.83e | — |
| Second-instar | 10 | 97.81 ± 1.98d | 35.93 ± 2.44c |
| Third-instar | 10 | 150.83 ± 4.39c | 53.02 ± 5.29b |
| Fourth-instar | 10 | 188.38 ± 20.69b | 57.97 ± 3.66b |
| Fifth-instar | 10 | 316.11 ± 3.22a | 107.51 ± 5.74a |

Means with same letters in the same column are not significantly different (GLM, Tukey, $P > 0.05$). '—', indicates absent.

**Table 2. Abundance and distribution of sensilla on the antennae of the five nymphal stages of *Diaphorina citri*.**

| Nymphal stages | | Sensilla | | | | | | | | | | Total | |
|---|---|---|---|---|---|---|---|---|---|---|---|---|---|
| | | ST | TH1 | TH2 | PSO | CvS1 | CvS2 | SCA | SB1 | SB2 | SB3 | | |
| First-instar | Se1 | — | — | — | — | — | — | — | — | — | — | — | 6 |
| | Se2 | 2 | 1 | 1 | 1 | 1 | — | — | — | — | — | 6 | |
| Second-instar | Se1 | 1 | — | — | — | — | — | — | — | — | — | 1 | 9 |
| | Se2 | 3 | 1 | 1 | 2 | 1 | — | — | — | — | — | 8 | |
| Third-instar | Se1 | 2 | — | — | — | — | — | — | — | — | — | 2 | 17 |
| | Se2 | 3 | 1 | 1 | 2 | 1 | 1 | — | 1 | 3 | 2 | 15 | |
| Fourth-instar | Se1 | 2 | — | — | — | — | — | 1 | — | — | — | 3 | 18 |
| | Se2 | — | — | — | — | — | — | — | 1 | 1 | — | 2 | |
| | Se3 | 4 | 1 | 1 | 3 | 1 | — | — | 1 | 2 | — | 13 | |
| Fifth-instar | Se1 | 5 | — | — | — | — | — | 1 | — | — | — | 6 | 31 |
| | Se2 | 3 | — | — | — | — | — | — | — | 3 | — | 6 | |
| | Se3 | 5 | 1 | 1 | 4 | 1 | 1 | — | 3 | 2 | 1 | 19 | |

Number and location of the various sensilla observed on the antennae for the five nymphal stages of *D. citri*. Se1, Se2 and Se3, the first, second and third segment of the antennae; ST, sensilla trichoidea; TH1 and TH2, the long and short terminal hair at the tip of the antennae, respectively; PSO, partitioned sensory organ; CvS1 and CvS2, cavity sensillum 1 and 2, respectively; SCA, sensilla campaniform; SB1, SB2 and SB3, sensilla basiconica 1, 2 and 3, respectively; '—', indicates absent.

socket (Fig 3), and gradually tapered to the apex of the antennae (Fig 3B). Only one ST positioned just below the base of the TH1 in each development stage of *D. citri* (Fig 3A).

**Partitioned sensory organ.** Partitioned sensory organ (PSO) was located on the ventral side of the last segment of the antennae of each nymphal stage of *D. citri* (Table 2). The PSO formed with an open pit and a sheet structure from which a cavity sensillum protruded (Fig 4A). The inner side of the PSO was enclosed by the cuticular fringe (Fig 4B). One large PSO was found in the first-instar nymphs, two in the second- and third-instar nymphs, three in the fourth-instar nymphs and four in the fifth-instar nymphs, respectively (Table 2).

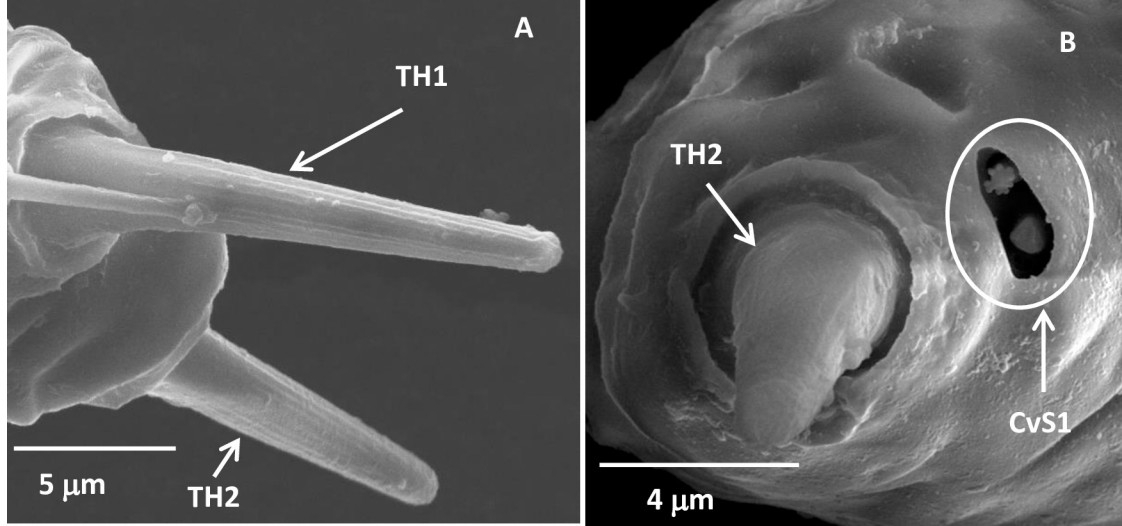

**Fig 2. The antennal tip of the third-instar *Diaphorina citri* nymphs.** (A) Two terminal hairs and sensilla trichoidea. (B) The short terminal hair and cavity sensillum 1. TH1, the long terminal hair; TH2, the short terminal hair; CvS1, cavity sensillum 1.

**Table 3. Length and width (mean ± SE, *n* = 10) of TH1 and TH2 of the five nymphal stages of *Diaphorina citri*.**

| Stages | Length (μm) | | Width(μm) | |
|---|---|---|---|---|
| | TH1 | TH2 | TH1 | TH2 |
| First-instar | 16.03±0.62ABa | 13.60±0.68ABb | 2.16±0.14Ca | 1.99±0.14Ca |
| Second-instar | 13.38±0.66Ba | 9.88±0.33Bb | 2.37±0.20Ca | 2.45±0.09Ca |
| Third-instar | 16.34±0.85ABa | 10.30±0.83Bb | 2.53±0.09Ca | 2.73±0.19BCa |
| Fourth-instar | 18.87±1.33ABa | 13.00±0.51ABb | 2.90±0.25BCa | 3.65±0.30Ba |
| Fifth-instar | 21.89±1.32Aa | 15.13±1.23Ab | 4.05±0.49ABa | 4.81±0.26Ab |

Means in columns with same uppercase letters are not significantly different (GLM, Tukey, *P*>0.05). Means in rows with same lowercase letters are not significantly different (*P*>0.05) in Mann–Whitney *U* test.

**Cavity sensillum.** Only one cavity sensillum 1 (CvS1) was found below the base of the TH2 in each nymphal stage of *D. citri* (Table 2). They were an oval sensory cavity formed by invaginations of the antennal cuticle and contained two sensilla (Fig 5A and S1 Fig). The cuticular parts of the two sensilla sitting on the bottom of the cavity were formed as short pegs, terminating just below the antennal surface. The tips of two short pegs were flat, and one of them was flower-shaped.

Cavity sensillum 2 (CvS2) was only found on the ventral side of the base of the third segment of the fifth-instar nymphal antennae (Table 2). Similar to the CvS1, they were a round sensory cavity containing only one sensilla peg with a flower-shaped tip (Fig 5B and 5C). The tip of the flower-shaped tip was not flat.

**Sensilla basiconica.** Sensilla basiconica 1 (SB1) was distributed on the dorsal side of the third-, fourth- and fifth-instar nymphal antennae (Table 2). The SB1 was fitted into a big socket elevated above the cuticle, measuring 6.35 μm in length and 1.73 μm in basal diameter. They displayed a longitudinally grooved surface with a sharp tip (Fig 6A).

Sensilla basiconica 2 (SB2), similar to the SB1, was also found on the dorsal side of the third-, fourth- and fifth-instar nymphal antennae (Table 2). They were robust on the antennae, grew in a big basal socket, and measured 31.07 μm in length and 3.31 μm in basal diameter.

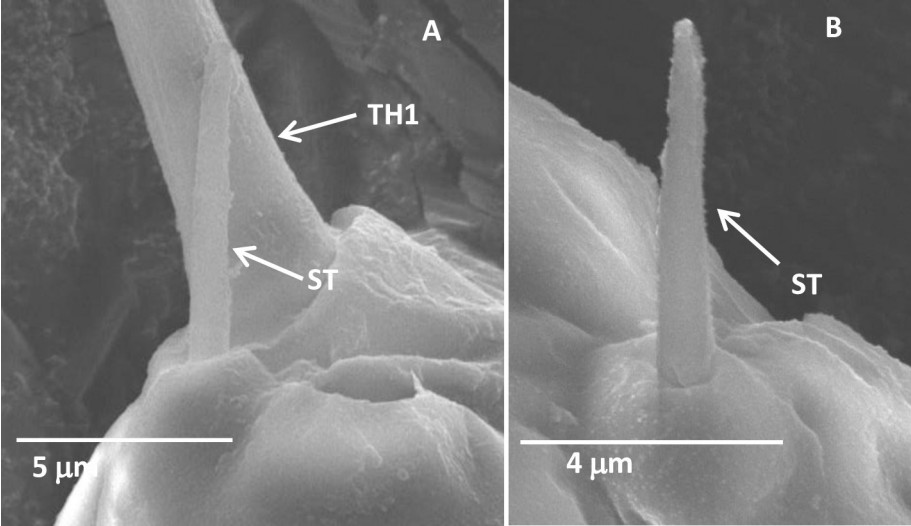

**Fig 3. Sensilla trichoidea in the third-instar *Diaphorina citri* nymphs.** (A) The long terminal hair and sensilla trichoidea. (B) Sensilla trichoidea. TH1, the long terminal hair; ST, sensilla trichoidea.

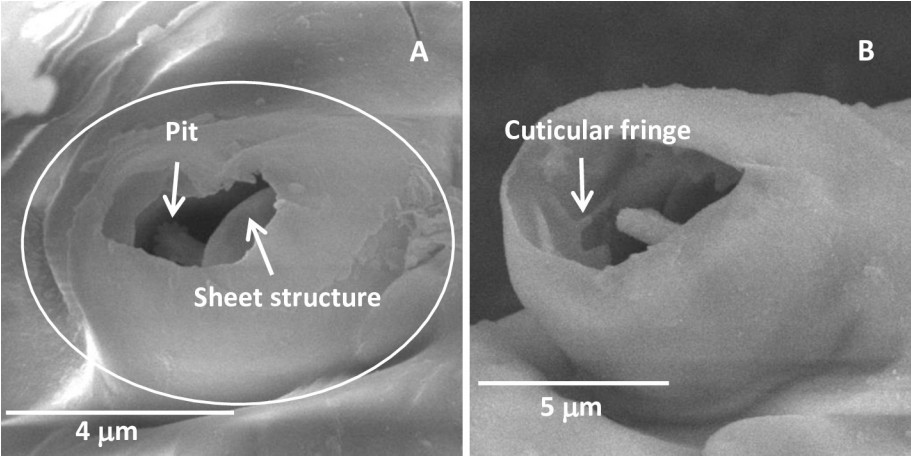

**Fig 4. Partitioned sensory organ in the fifth-instar *Diaphorina citri* nymphs.** (A) Partitioned sensory organ, showing the pit and sheet structure. (B) Partitioned sensory organ, showing the cuticular fringe.

The SB2 had multiple-pitted surfaces formed by clutter gullies, like the amygdala (Fig 6B). The number of SB2 was much more than that of the SB1 (Table 2).

Sensilla basiconica 3 (SB3) was sparsely distributed on the dorsal side of the last segment of the third- and fifth-instar nymphal antennae (Table 2). Two SB3 were found in the third-instar nymphs and one in the fifth-instar nymphs. The SB3 had a wavy line surface, like noodles (Fig 6C). They were also robust and grew in a big basal socket, measuring 12.7 μm in length and 3.86 μm in basal diameter.

**Sensilla campaniform.** A single sensilla campaniform (SCA) was found on the dorsal side of the second segment of the antennae of the fourth- and fifth-instar nymphs (Table 2 and Fig 7). The SCA had a dome-shaped sensory structure. Their ambient cuticle was protuberant but sunken in the middle (Fig 7A). In addition, there was a concave spot at the sunken structure in SCA (Fig 7B).

## Discussion

The morphology and ultrastructure of the antennae and antennal sensilla in different nymphal stages of *D. citri* were studied. The antennae of *D. citri* were composed of two segments in the

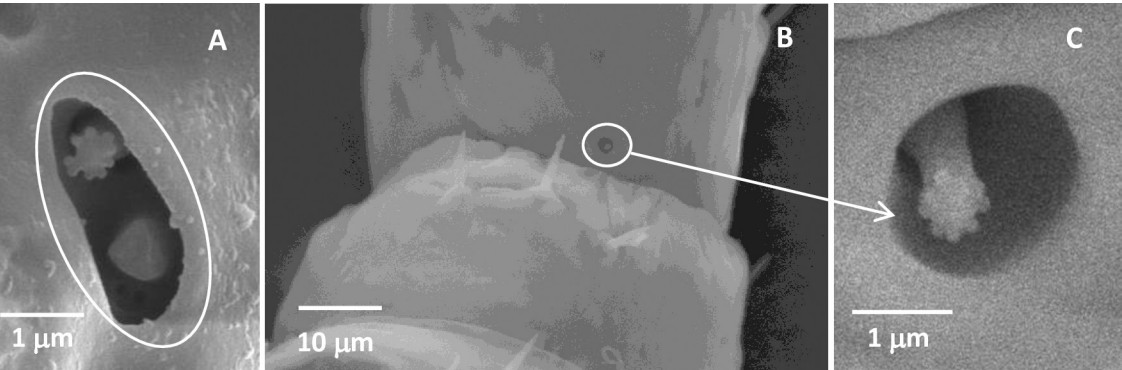

**Fig 5. Cavity sensillum in the fifth-instar *Diaphorina citri* nymphs.** (A) Cavity sensillum 1, containing an oval sensory cavity and two pegs. (B) Cavity sensillum 2, containing a round sensory cavity and one peg. (C) The high magnification picture of cavity sensillum 2.

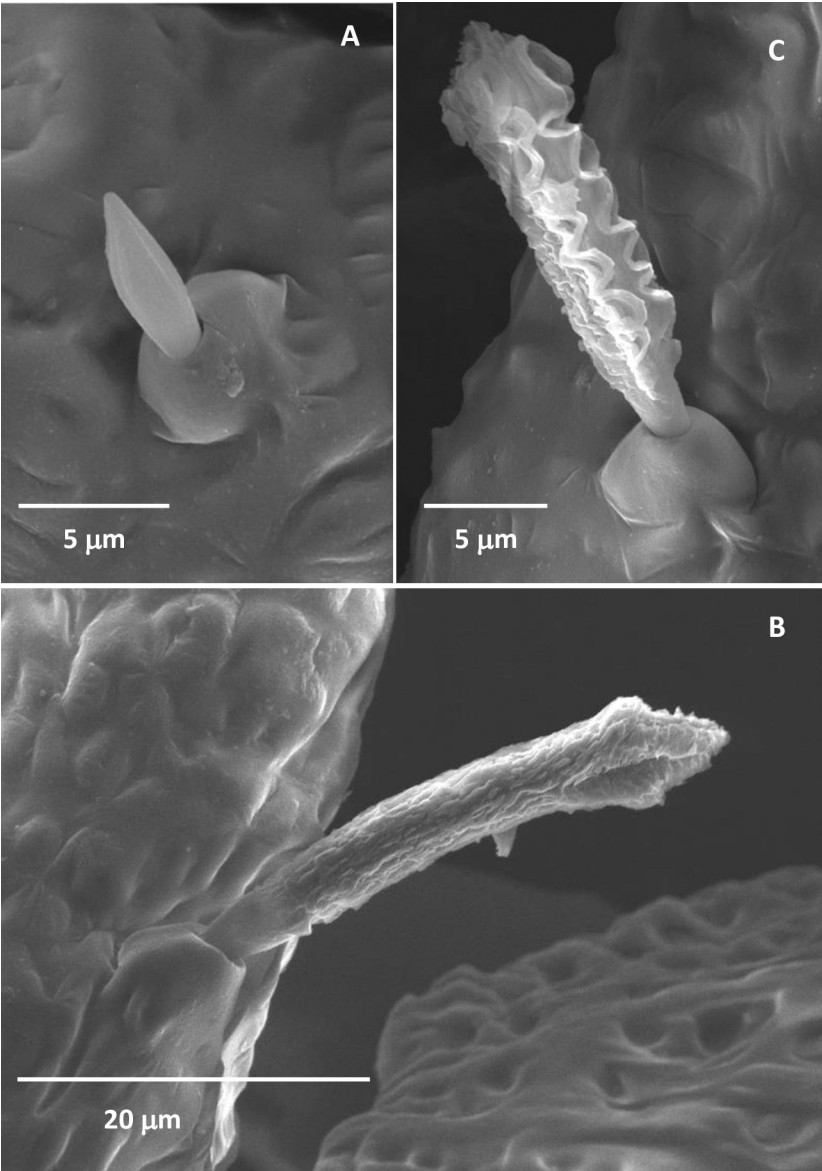

**Fig 6. Sensilla basiconica in the fourth-instar *Diaphorina citri* nymphs.** (A) Sensilla basiconica 1. (B) Sensilla basiconica 2. (C) Sensilla basiconica 3.

first-, second- and third-instar nymphs, and two basal segments and the flagellum with poorly defined subsegments in the fourth- and fifth-instar nymphs (Fig 1). Like Onagbola et al. [20], we did not find the SB1-3, CvS1 and CvS2 in adult *D. citri* (S1 File and S1 Table). Perhaps, they were degenerated and missed in adults. The PSO of the nymphal instars had a 'raised edges and internal shutter-like' appearance which might be replaced by the antennal rhinaria in adult *D. citri* (S1 File).

The THs were analogous with those found in *Psylla pyricola* Förster (Hemiptera: Psyllidae) [27], *Trialeurodes vaporariorum* (Westwood) (Hemiptera: Aleyrodidae), *Aleyrodes proletella* (Linnaeus) (Hemiptera: Aleyrodidae), *Bemisia tabaci* (Gennadius) (Hemiptera: Aleyrodidae) [30,33], *Trioza apicalis* (Hemiptera: Triozidae) [31], *Dialeurodes citri* (Ashmead) (Hemiptera: Aleyrodidae) [34], *Aleurodicus dispersus* Russell (Hemiptera: Aleyrodidae) [32,35]. As is the

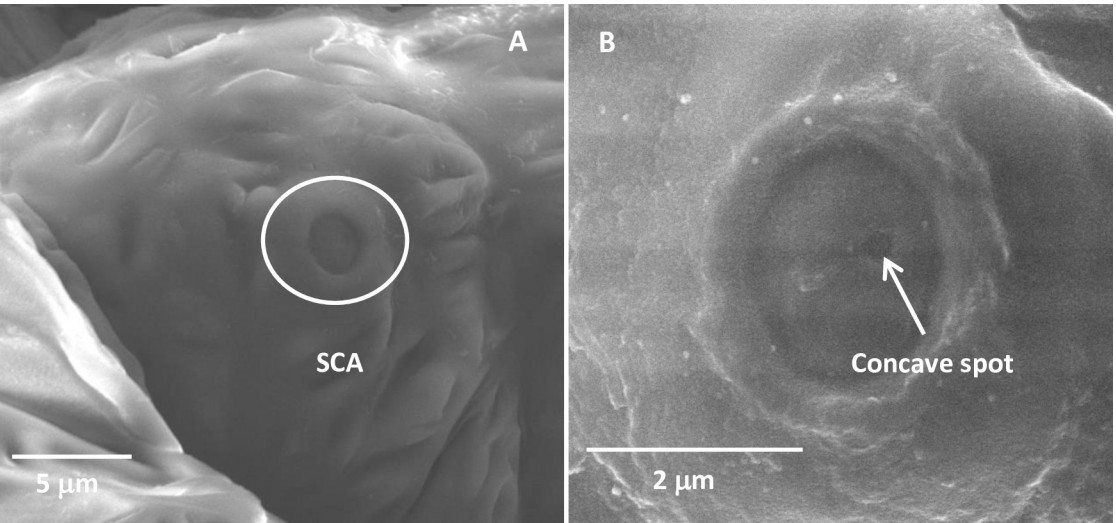

**Fig 7. Sensilla campaniform in fourth-instar *Diaphorina citri* nymphs.** (A) Sensilla campaniform on the second segment of the antennae. (B) Sensilla campaniform, showing the concave spot. SCA, sensilla campaniform.

case with other types of sensilla, the terminal hairs have been differently named by various authors. For example, Singleton-Smith et al. [27] and Mellor and Anderson [30] characterized them as "chaetica sensilla", which can be suggested to serve as a proprioreceptor perceiving antennal movement and position or have mechano-sensory functions [20,36], due to their location and presence to membranous connections between the sensillar shaft and the antennal cuticular surfaces [37]. Considering that the TH1 and TH2 in adults (Fig 2 in S1 File) have wall pores in adult antennae, it is conceivable that they are olfactory sensilla.

Sensilla trichoidea have been reported to perform either or both mechano- and chemo-sensory function [7]. The ST with no cellular material and pore canal structures (S2 Fig) resembled the mechanosensory hair which positioned just below the base of the TH1 in carrot psyllid [31]. In addition, the ST present in analogous locations with the AST-2 described by Onagbola et al. [20] may also suggest to have mechanosensory functions [20,31].

The PSO was first reported in *P. pyricola* and suggested to be chemoreceptors [27]. Their morphology, location, and number were similar in the five different nymphal instars between *P. pyricola* and *D. citri*. Previous studies have reported that the descriptions of the plate organs (sensilla placodea) in aphids conformed closest to the large sensory organ (PSO) [25,38–40]. Therefore, if these were indeed plate organs that were placed in the category of chemoreceptor by Slifer [40], they could be involved in olfactory function.

The CvSs were the compound sensilla which consisted of a sensory cavity and the sensilla pegs, but they were only meant the sensory cavity named by Kristoffersen et al. [31] and Onagbola et al. [20] in psyllids. The difference between CvS1 and CvS2 was the number of pegs in the cavity. All of their pegs were well hidden beneath the antennal surface, and their position was far from optimal for receiving chemical stimuli. Kristoffersen et al. [31] have reported that these pegs are likely to be a necessary adaptation to prevent desiccation. Moreover, several studies have demonstrated that sensilla that are recessed from the antennal surface and located within cavities may be involved in perception of the $CO_2$, humidity, and temperature [41–43]. Judging by morphological features, the CvS1 and CvS2 could be responsible for the perception of any type of stimuli as described above.

The SBs only occurred on the antennae of third-, fourth- and fifth-instar nymphs. Similarly, they were not reported in other nymphal or adult psyllids [20,27,31]. In our study, the SBs

were inserted into a big inflexible socket and exhibited different morphological surfaces (Fig 6). Judging by the morphological surface features, three subtypes of this sensillum can be identified. According to Schneider [25], sensilla basiconica are the most common and important chemoreceptors found on the antennae of insects. Moreover, Altner and Prillinger [44] and Zacharuk [45] suggested that it would be impossible for sensilla with an inflexible socket to act as a mechanoreceptor.

The SCA has been observed in many insects, including whiteflies [36] and psyllids [27]. On the antennae of the fourth- and fifth-instar nymphs, only one SCA was presented on the second segment of the antennae. The SCA seems to occur in aging nymphs (the fourth-instar nymph of *A. dispersus* and the fifth-instar nymph of *P. pyricola*) [27,32] or adults (*Tetrastichus howardi* and *A. dispersus*) [35,46]. Previous studies have reported that the SCA with no pores in their cuticular structure play the role of mechanoreceptors [29,37,47–51], whereas the SCA with pores was involved in gustatory system as well as in humidity reception [52]. Here they probably acted as mechanoreceptors, because they were few and located near the segmental joints [7,26,53,54].

In our study, the types and total number of the antennal sensilla obviously increased in the third-, fourth- and fifth-instar nymphs. According to Hung et al. [55], *C.* Liberibacter asiaticus (Las) persists in the ACP vector but is not transovarially transmitted by the vector. The pathogen can only be transmitted by the third-, fourth- and fifth-instar nymphs and adults which play a major role in the spread of the pathogen because of their dispersal capabilities [2,55–57], but the nymphs were found to be more efficient [4]. Further studies are needed to determine whether the antennal sensilla in the third-, fourth- and fifth-instar nymphs (especially the SBs) are related to the feeding behavior of nymphs on Las-infected citrus. In conclusion, we have provided an extensive description of the antennae and antennal sensilla in different nymphal stages of *D. citri* using SEM. This information can be a great help for revealing the developmental course of psyllid's antennae and antennal sensilla and provides new insights on the olfactory behavior of *D. citri* nymphs.

## Supporting information

**S1 Fig. Transmission electron microscopy micrograph of the transect through the cavity sensillum 1 of the first-instar *Diaphorina citri*, showing the two sensilla in the cavity (arrows).**
(DOC)

**S2 Fig. Transmission electron microscopy micrograph of the sensilla trichoidea positioned below the base of the TH1 in the third-instar *Diaphorina citri* nymphs.**
(DOCX)

**S1 Table. Abundance and distribution of sensilla on the antennae of adult *Diaphorina citri*.**
(DOCX)

**S1 File. General analysis of the antennae and antennal sensilla of adult *Diaphorian citri*.**
(DOC)

## Acknowledgments

The authors are grateful to Wei-Chun Li from Jiangxi Agricultural University for English revision and critical reading of the manuscript. We would like to thank Ms Xiao-Yin Hu from the Centre Laboratory of the South China Botanical Garden for her valuable assistance with the

treatment for the specimens in SEM. Special thanks to Wei-Jian Wu from South China Agricultural University for the statistical analysis.

## Author Contributions

**Conceptualization:** Lixia Zheng.

**Data curation:** Lixia Zheng, Qichun Liang, Wensheng Chen.

**Formal analysis:** Lixia Zheng.

**Funding acquisition:** Lixia Zheng.

**Investigation:** Lixia Zheng.

**Methodology:** Lixia Zheng, Wensheng Chen.

**Supervision:** Lixia Zheng, Wensheng Chen.

**Writing – original draft:** Lixia Zheng, Wensheng Chen.

**Writing – review & editing:** Lixia Zheng, Ming Yu, Yi Cao, Wensheng Chen.

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
