## [Editor Report · Decision Letter 0]

4 Oct 2019

PONE-D-19-23769

Morphological characterization of nymphal antennae and antennal sensilla of Diaphorina citri Kuwayama (Hemiptera: Psyllidae)

PLOS ONE

Dear Dr. Zheng,

Thank you for submitting your manuscript to PLOS ONE. After careful consideration, we feel that your manuscript needs additional work before it is considered for peer review. Therefore, we invite you to submit a revised version of the manuscript that addresses the points raised during the review process.

I appreciate your effort to write in a second language and the improvements you made in the revised version, but there is still work to be done:

(1) Change into black all red letters left in the m/s; (2) send the m/s for a second cycle of editorial processing; there are many errors, such as DiscuRsion, flower shape vs. flower shaped, The PSO formed with an opened..., the first segment of the antennae was absent sensilla, citrus psylla (the official common name for D. citri is Asian Citrus Psyllid (ACP), Acknowledge, to cite a few (please have a thorough review); (3) the literature is outdated; to a minimum you have to cite the work of Illiano Coutinho when referring to D. citri antennal structure; our two papers on pheromones (please perform a in-depth literature search and update your references)_; (4) follow PLoS ONE instructions (Tables go in the Main Text); (5) improve figure legends; some of them just have a title; (6) bring relevant TEM figures to the main text. 

I am taking the unusual step of offering you suggestions for improvement. Here, I just pointed out a few issues I observed while evaluating the m/s, but please make certain that you perform a really thorough processing of the manuscript before re-submission.

We would appreciate receiving your revised manuscript by Nov 18 2019 11:59PM. To enhance the reproducibility of your results, we recommend that if applicable you deposit your laboratory protocols in protocols.io, where a protocol can be assigned its own identifier (DOI) such that it can be cited independently in the future. For instructions see: http://journals.plos.org/plosone/s/submission-guidelines#loc-laboratory-protocols

We look forward to receiving your revised manuscript.

Kind regards,

Walter S. Leal

Academic Editor

PLOS ONE

**Journal Requirement**s

2.Please include captions for your Supporting Information files at the end of your manuscript, and update any in-text citations to match accordingly. Please see our Supporting Information guidelines for more information: http://journals.plos.org/plosone/s/supporting-information.

---

## [Author Response · Author response to Decision Letter 0]

28 Oct 2019

We have studied the comments carefully and have made correction which we hope meet with approval. The main corrections in the paper and the responds to the comments are as flowing:

Responds:

1 Change into black all red letters left in the m/s.

Response: Thank you very much. We have done and checked several times.

2 send the m/s for a second cycle of editorial processing; there are many errors, such as DiscuRsion, flower shape vs. flower shaped, The PSO formed with an opened..., the first segment of the antennae was absent sensilla, citrus psylla (the official common name for D. citri is Asian Citrus Psyllid (ACP), Acknowledge, to cite a few (please have a thorough review).

Response: We are grateful to Professor Walter S. Leal for the nice and good comments. We are very sorry for the errors which should not be made. So, we have corrected the errors not only listed by Professor Walter S. Leal, but also revised some other errors. Revised portion can be found in the file “Revised Manuscript with Track Changes” easily.

3 the literature is outdated; to a minimum you have to cite the work of Illiano Coutinho when referring to D. citri antennal structure; our two papers on pheromones (please perform an in-depth literature search and update your references).

Response: Thank you very much for the good suggestions. In fact, we have read these two references (“Zanardi OZ, Volpe HXL, Favaris AP, Silva WD, Luvizotto RAG, Magnani RF, et al. Putative sex pheromone of the Asian citrus psyllid, Diaphorina citri, breaks down into an attranctant. Scientific Reports. 2018; 8(1): 455.”; “Laboratory and field evaluation of acetic acid-based lures for male Asian citrus psyllid, Diaphorina citri. Scientific Reports. 2019; 9(1): 1-10.”), but we do not know how to refer. Our teams also work for the sex pheromone of the ACP. You can find these new references in the page 2 lines55-59. 

4 Follow PLoS ONE instructions (Tables go in the Main Text).

Response: Done. Thank you!

5 Improve figure legends; some of them just have a title.

Response: Done. Thank you!

6 Bring relevant TEM figures to the main text. 

Response: Thank you very much for the suggestion. However, the TEM figures of the five different nymphal stages are very poor. We do think they can meet the requirements of PLoS One.

We tried our best to improve the manuscript and made some changes in the manuscript. We appreciate for warm work of Professor Walter S. Leal earnestly, and hope that the correction will meet with approval.

Once again, thank you very much for your comments and suggestions.

---

## [Decision Letter · Decision Letter 1]

27 Nov 2019

PONE-D-19-23769R1

Morphological characterization of nymphal antennae and antennal sensilla of Diaphorina citri Kuwayama (Hemiptera: Psyllidae)

PLOS ONE

Dear Dr. Zheng,

Thank you for submitting your manuscript to PLOS ONE. After careful consideration, we feel that it does not fully meet PLOS ONE’s publication criteria as it currently stands. Therefore, we invite you to submit a revised version of the manuscript that addresses the points raised during the review process.

Please note that none of the reviewers was favorable to publication in PLoS ONE. I would like, however, to give you the opportunity to address the reviewers' concerns. Reviewer #2 suggested that adding TEM would raise the level of your manuscript. All three reviewers provided constructive criticism that you should address most carefully. Once you have addressed all concerns and are almost ready to re-submit (if you so decide), please make certain you have the manuscript edited before resubmission.

We would appreciate receiving your revised manuscript by Jan 11 2020 11:59PM. Because your manuscript requires extensive work, we would be willing to grant you an extension of this deadline, if needed. To enhance the reproducibility of your results, we recommend that if applicable you deposit your laboratory protocols in protocols.io, where a protocol can be assigned its own identifier (DOI) such that it can be cited independently in the future. For instructions see: http://journals.plos.org/plosone/s/submission-guidelines#loc-laboratory-protocols

We look forward to receiving your revised manuscript.

Kind regards,

Walter S. Leal

Academic Editor

PLOS ONE

Reviewers' comments:

Reviewer's Responses to Questions

**Comments to the Author**

1. If the authors have adequately addressed your comments raised in a previous round of review and you feel that this manuscript is now acceptable for publication, you may indicate that here to bypass the “Comments to the Author” section, enter your conflict of interest statement in the “Confidential to Editor” section, and submit your "Accept" recommendation.

Reviewer #1: (No Response)

Reviewer #2: (No Response)

Reviewer #3: (No Response)

2. Is the manuscript technically sound, and do the data support the conclusions?

Reviewer #1: Yes

Reviewer #2: Partly

Reviewer #3: Partly

3. Has the statistical analysis been performed appropriately and rigorously? 

Reviewer #1: Yes

Reviewer #2: Yes

Reviewer #3: I Don't Know

4. Have the authors made all data underlying the findings in their manuscript fully available?

Reviewer #1: Yes

Reviewer #2: Yes

Reviewer #3: Yes

5. Is the manuscript presented in an intelligible fashion and written in standard English?

Reviewer #1: Yes

Reviewer #2: No

Reviewer #3: No

6. Review Comments to the Author

Reviewer #1: (No Response)

Reviewer #2: In this manuscript, Zheng and Chen employed scanning electron microscopy (SEM) to extensively investigate the external anatomy of the antenna of one of the important pests of citrus trees, a psyllid, Diaphorina citri. The authors made a detailed comparison of the antennal sensilla across the five nymphal instars, the most efficient transmitters of a bacterium that causes citrus greening disease. They showed that the antennae of the first- through third- nymphal instars contain two flagellomeres whereas those of fourth- and fifth-nymphal instars possess three flagellomeres. The authors identified 11 distinct antennal sensilla types spread almost uniformly over the antennae except on the first segment. Studies of this kind are essential to future electrophysiological, behavioral and molecular investigations on this insect pest. However, I have three major concerns and some minor comments:

Major concerns:

1- The manuscript suffers from the lack of an appropriate structural origination. Specifically, in the results section, while the first part focuses mainly on the nymphal instars the authors also deal solely with the presence (or absence) and distribution of various types of the sensilla without first presenting their characteristics. This definitely confuses the readers. I’d suggest the authors rewrite the results section by combining the second part of the results i.e. “Morphology and Structure of Sensilla” with the first part so that in each paragraph the description of each sensilla type should be followed by its presence (or absence), number, distribution, etc., in each nymphal stage. The authors are also encouraged to better formulate their hypothesis in the introduction (the information is there but is not very well promoted!).

2- Lack of TEM and/or electrophysiological studies: Usually studies of this kind are accompanied by TEM and/or electrophysiological experiments for strengthening the results. I understand that due to, e.g. lack of equipment or budget constraints, it might not be feasible to conduct the above-mentioned experiments. Hence, I’d suggest that …

3- … the authors submit their findings to other journals (such as Micron) that publish these kinds of works so as to attract more readers interested in the SEM.

Minor comments:

I believe the following items will increase the value of the manuscript:

1- Inclusion of parameters like the size of individual flagellomeres across the nymphal instars

2- Figuring out if there is a correlation between the nymph size within each nymphal stage and the number of sensilla?

3- Were the individuals size-matched? I’d suggest the authors include the average head-width (indicative of size of the nymphs) of each nymphal stage in table 1.

• Lines 14-15: “Because its ability to transmit C. Liberibacter was more efficient in nymphs” please change it to “Because the nymphal instars have been reported to more efficiently transmit C. Liberibacter”.

• Line 25: “The distribution of antennal sensilla in each nymphal stage of D. citri was asymmetrical”. Do you mean they are “randomly distributed”?

• Lines 93-97: Under statistical analysis, instead of clearly explaining how they analyzed their data, the authors refer their readers to 11 references that focus on the classification of the sensilla. Please remove the author names and instead write, “…similar structures described previously [27-35].

• When the antennae are mounted only one side of the antennae is facing upward. It is not clear how the number of different sensilla were counted. Please elaborate on that.

• Figure legends must be self-explanatory. For example in figure 1 it must be clearly explained that A-C belong to the 1st through the 3rd nymphal instars with two antennal flagellomeres and D-E belongs to the 4th and 5th nymphal with three flagellomeres.

• Lines 139-140: basal segments and subsegments must be indicated in the figure.

• “Se 1” or “S1”? Please be consistent.

• Line 244: please change “trichoids hairs” to “trichoid sensilla”

Reviewer #3: Data partialy supports discussion and conclusions: the authors are unable to explain the absence of some sensillas in adults, since they did not assess adult antenna structures.

I am not sure about the use/choice (parametrical/non-parametrical analyses) because the authors did not mention if data normality was tested, neither number of replicates.

There are some mispelling in the text (comments and corrections in the pdf file)

. The species family is wrong. Liviidae instead Psyllidae

7. PLOS authors have the option to publish the peer review history of their article (what does this mean?). If published, this will include your full peer review and any attached files.

Reviewer #1: No

Reviewer #2: No

Reviewer #3: No

---

## [Author Response · Author response to Decision Letter 1]

7 Jan 2020

We have studied the comments carefully and have made correction which we hope meet with approval. The main corrections in the paper and the responds to the comments are as flowing:

Responds to Reviewer 1 (file “review”):

Zheng et al observed the antennae of the Diaphorina citri nymph, and identified the sensilla on the antennae with scanning electron microscopy. The general morphology difference was clarified in different nymph stages. The number, distribution, ultrastructure of the sensilla (11 types) on the nymph antennae was recorded. The function of these sensilla was deduced by comparing with the sensilla of other insect species to explore their role of in Candidatus Liberibacter transmitting and D. citri nymph feeding. The results is very interesting. However, this manuscript need some improvements to publish, especially the quality of the photo. It is difficult to get useful information without clear photo showing the characteristic of the sensilla.

1 in the method section of page 3 (line 82-83), why the author used the double-sided adhesive tape, but not a carbon conductive tape. In line 84, the author said they used a XL30, and Nova Nano 430 scanning electron microscopy at 10 kV. In figure 5,B, the photo labeled with 25kV.

Response: Thank you very much for your carefulness. We have performed the studies of SEM both at South China Agriculture University and Centre Laboratory of the South China Botanical Garden several times, but the treatment method and process of the specimens are the same. Most of the SEM photos were got at South China Agriculture University using an SEM (XL30, FEI, Holland and Nova Nano 430, FEI, Holland) at 10kV, few SEM photos were got at Centre Laboratory of the South China Botanical Garden using an SEM (JSM-6360LV, Japan) at 25kV. The reason for the specimens were used the double-sibed adhesive tape, actually, we don’t know why. The method of anchoring the specimens was provided by the researchers in these two electron microscope laboratories.

2 in line 94, page 4, the length and width of various stages of long terminal hair and short terminal hair were determined. How they measured the length and the width. How many insects they measured. How to minimize the measurement differences due to the viewing angles.

Response: Thank you very much for your comments. The length and width (basal diameter) of the long terminal hair and short terminal hair are measured from the printed SEM images by a slide caliper. The starting and ending points of the measured data are all in the middle. Ten insects of each stage of D. citri were examined.

3 I suggest to mark the segments of the antennae in figure 1. For example, it is difficult to find any sensillum on the second- instar nymph without any mark in figure 1A (line 115) and 1B (line 122). 

Response: Thank you very much for your suggestion. The first and second segments of the antennae in five different nymphal stages are segmented clearly, and we have marked for the fourth-instar in Fig 1D. In the first- and second-instar nymphs, the total number of antennal sensilla is very small; moreover, they were distributed either on the ventral side or on the dorsal of the antennae. So, we cannot observe all of them in one picture.

4 All the photos are blurry (especially the photo taken by Nova Nano 430) and it is recommended to shoot again. A good result maybe obtain by Nova Nano 430 at low voltage (for example 5Kv) without dehydrating and drying of the insect. 

Response: Thank you very much for your suggestion. In order to get the high quality photos, we performed our SEM study both at South China Agriculture University and Centre Laboratory of the South China Botanical Garden several times. We even tried to take the specimens without any treatments to anchor for examining by SEM. Here, we hope our photos may meet the requirement of PLoS One.

5 What is the difference between terminal hair and TH1 and TH2 (in paragraph 2, page 6). How to determine the TH1 is longer than TH2, rather than the visual illusion caused by view angle.

Response: Thank you very much for your comments. TH1 was distributed with a ST on the antennal tip of different instars of D. citri (see Fig 3), including adult females and males (see Fig 1 and Fig 2A in S1 File). In D. citri nymphs, we don’t find the morphological difference between TH1 and TH2 except the length and basal diameter (width). However, there is the obvious difference found in adults (see Fig 3 in S1 File). Moreover, the length of TH1 and TH2 was significant difference in the length and width (see Table 2 in S1 File).

6 What is the last ill-defined segment and partitioned sensory organ? (line 171,page 7). A concise legend is needed for figure 4 to help it stand alone ( useful information can be found here, https://blog.bioturing.com/2018/05/10/how-to-craft-a-figure-legend-for-scientific-papers/)

Response: Thank you very much for your comments. Partitioned sensory organ was located on the last segment of the antennae of each ACP nymphal stage. The fourth- and fifth-instar nymphs had an ill-defined flagellum which was also the last segment of the antennae. Our expression is not very accurate, and we have revised it. Also, we have checked all of our figure legends, and thank you very much.

7 What is the cavity sensillum, where it come from?

Response: Thank you very much for your comment. Cavity sensillum consisted of a sensory cavity and one (Fig 5B) or two pegs (Fig 5A). I have marked the cavity sensillum with a circle, maybe it can be easy to understand. I am very sorry that I don’t really understand that “where it come from?” All sensilla in our study were distributed on the antennae of five different nymphal stages of D. citri.

8 In the discussion section, paragraph 2, the author claimed that another name of TH is chaetica sensilla, why?

Response: In most of whiteflies and psyllids such as Trialeurodes vaporariorum, Aleyrodes proletella, Bemisia tabaci, Dialeurodes citri, Aleurodicus disperses, Psylla pyricola, Trioza apicalis and D. citri, there are one or two TH found on the antennal tip. According to Singleton-Smith et al. (1978) and Mellor and Anderson (1995), they are characterized as “chaetica sensilla”, maybe due to their location and the morphology of that they are obvious strong and long with longitudinal groove surface and inserted into a big cuticular socket. These antennal sensilla should be the first to touch and perceive the external environment because of their location and size.

(Singleton-Smith J, Chang JF, Philogène JR. Morphological differences between nymphal instars and descriptions of the antennal sensory structures of the nymphs and adults of Psylla pyricola Foerster (Homoptera: Psyllidae). Can J Zool. 1978; 56(7): 1576-1584.)

(Mellor HE, Anderson M. Antennal sensilla of whiteflies: Trialeurodes vaporariorum (Westwood), the glasshouse whitefly, Aleyrodes proletella (Linnaeus), the cabbage whitefly, and Bemisia tabaci (Gennadius), the tobacco whitefly (Homoptera: Aleyrodidae). Part 1: external morphology. International Journal of Insect Morphology and Embryology. 1995; 24(2): 133-143.)

9 The reference format should be checked carefully, and keep them in consistent. 2,3, 5,17, 18,,24,37,39,41 44,

Response: Done, thank you very much.

Special thanks to you for your comments and suggestions.

Responds to Reviewer 2 (from the decision letter):

1 The manuscript suffers from the lack of an appropriate structural origination. Specifically, in the results section, while the first part focuses mainly on the nymphal instars the authors also deal solely with the presence (or absence) and distribution of various types of the sensilla without first presenting their characteristics. This definitely confuses the readers. I’d suggest the authors rewrite the results section by combining the second part of the results i.e. “Morphology and Structure of Sensilla” with the first part so that in each paragraph the description of each sensilla type should be followed by its presence (or absence), number, distribution, etc., in each nymphal stage. The authors are also encouraged to better formulate their hypothesis in the introduction (the information is there but is not very well promoted!).

Response: Thank you very much for your suggestion. We tried to revise our MS according to your suggestion, but we found a little messy after adjusting the part of ‘Result’ of MS layout. We have revised our MS carefully according to the comments, changed the argument used to justify the choice for nymphal stages of D. citri, and added some description to improve the structure and quality of our MS, such as lines 147-153.

2 Lack of TEM and/or electrophysiological studies: Usually studies of this kind are accompanied by TEM and/or electrophysiological experiments for strengthening the results. I understand that due to, e.g. lack of equipment or budget constraints, it might not be feasible to conduct the above-mentioned experiments. Hence, I’d suggest that …

Response: Thank you very much for your good suggestion, we also try to get some high quality TEM photographs. From March 2017 to September 2018, we have performed the studies of SEM and TEM at South China Agriculture University and Centre Laboratory of the South China Botanical Garden, moreover, at Institute of Food Science and Technology CAAS for TEM. Unfortunately, we did not get good TEM photographs of different nymphal stages of Diaphorina citri. For the electrophysiological studies of D. citri nymphs, we have not plan to do, but we have done the adult electrophysiological studies. ACP is a small insect, and it is not easy to connect the antennae in the adult electrophysiological studies. Moreover, the various nymphal stages of D. citri antennae are much shorter than the adults (see Table 1 in S1 File), especially the first-, second-, third- and fourth-instar nymphal antennae (Table 1). Thank you very much for your suggestion, it is very helpful for our future researches.

3 … the authors submit their findings to other journals (such as Micron) that publish these kinds of works so as to attract more readers interested in the SEM.

Response: Thank you very much for your suggestion, most of our teams’ works about the SEM or TEM have published in Micron, Microscopy Research and Technique, as well as Plos One. ACP is the most important insect, and there are also quite a few researches about the ACP published in Plos One. We also think it is appropriate to submit our MS to Plos One.

4 I believe the following items will increase the value of the manuscript: 1) Inclusion of parameters like the size of individual flagellomeres across the nymphal instars. 2) Figuring out if there is a correlation between the nymph size within each nymphal stage and the number of sensilla? 3) Were the individuals size-matched? I’d suggest the authors include the average head-width (indicative of size of the nymphs) of each nymphal stage in table 1.

Response: 1) Thank you very much for your suggestions. It is very valuable and helpful for the important guiding significance to our future researches. 2) The length of the five different nymphal antennae was significantly increased with the increase of the nymphal instars, as well as the number of sensilla. We do not find the correlation between the nymph size within each nymphal stage and the number of sensilla. 3) Yes, it is size-matched. The classification characteristics of different stages of ACP nymphs are according to Ruan et al. (2012). The SEM photographs were just only antennae, not containing the whole head, so, it could not measure the ACP head-width of each nymphal stage. Thank you very much for your suggestions.

(Ruan CQ, Chen JL, Liu B, Duan YP, Xia YL. 2012. Morphology and behavior of Asian citrus psyllid, Diaphorina citri Kuwayama. Chinese Agricultural Science Bulletin, 28(31): 186-190)

5 Lines 14-15: “Because its ability to transmit C. Liberibacter was more efficient in nymphs” please change it to “Because the nymphal instars have been reported to more efficiently transmit C. Liberibacter”.

Response: Done, thank you very much.

6 Line 25: “The distribution of antennal sensilla in each nymphal stage of D. citri was asymmetrical”. Do you mean they are “randomly distributed”?

Response: No, it just means that the distribution of antennal sensilla in nymphs was not the same. The SBs, CvS2 and SCA are the examples of how the antennal sensilla in nymphs asymmetrically distribute. These sensilla are not distributed in each nymphal stage, only in the older nymphs such as third-, fourth- and fifth-instar.

7 Lines 93-97: Under statistical analysis, instead of clearly explaining how they analyzed their data, the authors refer their readers to 11 references that focus on the classification of the sensilla. Please remove the author names and instead write, “…similar structures described previously [27-35].

Response: Done, thank you very much.

8 When the antennae are mounted only one side of the antennae is facing upward. It is not clear how the number of different sensilla were counted. Please elaborate on that.

Response: Done, thank you very much. Each side of the antennae (dorsal, ventral, and two lateral surfaces of different nymphal stages of D. citri) was mounted to obverse. Each measured parameters was ten, and we have added the relevant contents in lines 90-92.

9 Figure legends must be self-explanatory. For example in figure 1 it must be clearly explained that A-C belong to the 1st through the 3rd nymphal instars with two antennal flagellomeres and D-E belongs to the 4th and 5th nymphal with three flagellomeres.

Response: Done, thank you very much.

10 Lines 139-140: basal segments and subsegments must be indicated in the figure.

Response: Done, thank you very much.

11 “Se 1” or “S1”? Please be consistent.

Response: I am very sorry for using such similar abbreviation. Actually, “Se1” is the abbreviation of the first segment of antennae, while “S1” is the abbreviation of the “S1 Fig.” in the file “Supplementary data”. We have renamed the supplementary.

12 Line 244: please change “trichoids hairs” to “trichoid sensilla”

Response: Done, thank you very much.

Special thanks to you for your comments and suggestions.

Responds to Reviewer 3 (from the decision letter):

1 Data partialy supports discussion and conclusions: the authors are unable to explain the absence of some sensillas in adults, since they did not assess adult antenna structures. 

Response: Thank you very much. Actually, we have performed the studies of adult D. citri with SEM and TEM. When we sent our first MS to same professors which have rich experience in SEM and TEM, they also advised us to delete the research on the adult D. citri because of the similar research reporting by Onagbola et al. (2008), We agree with the opinion, and keep some adult results in the part of ‘Discussion’ as supporting information (S1 File and S1 Table,). In adults, eight morphologically distinct types of sensilla were found to distribute on the antennae of female and male D. citri in our study (S1 File), with seven types reminiscent of those described by Onagbola et al. (2008). The aporous sensilla trichoidea occurred on the medial portions of the scape, pedicel and flagellomeres 1, 2, 4 and 5, the chaetica sensilla and the unidentified uniporous sensilla which were reported by Onagbola et al. (2008) were absent while the SCA were found in our study (S1 File). In our study, we do not descript the morphology and structure of antennae and antennal sensilla of adult D. citri.

(Onagbola EO, Meyer WL, Boina DR, Stelinski LL. Morphological characterization of the antennal sensilla of the Asian citrus psyllid, Diaphorina citri Kuwayama (Hemiptera: Psyllidae) with reference to their probable functions. Micron. 2008; 39(8): 1184-1191).

2 I am not sure about the use/choice (parametrical/non-parametrical analyses) because the authors did not mention if data normality was tested, neither number of replicates.

Response: Thank you very much. For the statistical analysis, we consulted an expert (Prof. Wei-Jian Wu from South China Agricultural University who are also major in Biostatistics) and we also added the relevant contents in lines 90-92.

3 There are some mispelling in the text (comments and corrections in the pdf file)

Response: Thank you very much for your comments, we have corrected according to your kind comments, and you can find the one to one response to the pdf file bellow.

4 The species family is wrong. Liviidae instead Psyllidae 

Response: Done. Thank you very much.

Responds to Reviewer 3 from the PDF file (PONE-D-19-23769_R1-reviewer):

1 The species was re-classified as Liviidae.

Response: Done. Thank you very much.

2 Line 12, “the most” instead of “an”.

Response: Done. Thank you very much.

3 Please re-phrase the sentence: The literature supports that nymphs were more efficient to transimit Ca. Liberibacter to the host plant. Maybe the fact can be related to feeding behavior aspects. However, nymphs presents capacity to move. On the other hands, adults can move from host plant to another one, transmiting the bacterium to health plants. Other aspect is related to chemical behavior. Initially, adults prefers Ca. Liberibacter plants (high MeSa emittion) and can acquire the bacterium, however, after some days, the adults move to health plants (better nutritional qualities). So, I am not sure that nymphs show a better ability to transmit when compared to adults.

Response: Done. Thank you very much. As reviewers’ comments and suggestions, it is inappropriate to justify the choice for immature stages. We have deleted the sentences, and added some discussion about these issues in the “Discussion” section.

4 I am not sure about the relation of the ‘ability’ of nymphs to transmit the bacterium (feed aspects) with the choice of nymphal antenna to perform the experiments. Since antenna is related with oddorants, and all behavioral studies with ACP assessed adults behavior and role of compounds, why the authors excluded adult antenna? This study is very welcome to complement ACP behavioral studies.

Response: Thank you very much for your kind suggestion, it is very valuable and helpful for the important guiding significance to our future researches. We have changed the sentence according to the comments. Actually, we also finished the study of adult D. citri, but they were not written in our MS because of the similar research reporting by Onagbola et al. (2008). So, we decide to present our adult ACP results as supplementary data in S1 File and S1 Table.

(Onagbola EO, Meyer WL, Boina DR, Stelinski LL. Morphological characterization of the antennal sensilla of the Asian citrus psyllid, Diaphorina citri Kuwayama (Hemiptera: Psyllidae), with reference to their probable functions. Micron. 2008; 39(8): 1184-1191.)

5 It is not clear for me what you would like to tell about number of antennae segments. There are 3 segments in insect antennae (scape, pedicel and flagellum). Please clarify what you mean with ‘structures’.

Response: Thank you very much for your comment. Yes, there are 3 segments in adult insect antennae (scape, pedicel and flagellum) generally. However, in some low-aged development nymphal stages, such as Aleurodicus dispersus and D. citri, their flagellum has not yet developed. In our study, we observed two segments of the antennae in first-, second- and third-instar of D. citri nymphs. In other words, their flagellum doesn’t develop in these ACP nymphs.

6 How many SBs for the other instar? Please clarify? Same for CvS2 and SCA. Please, you need to be clear about the number of sensilla (and sensilla types) that differ among the instars.

Response: Thank you very much for your suggestion. The details of SBs, CvSs and SCA are elaborated in the “Result” section, the number, abundance and distribution of sensilla are listed in Table 2. We don’t list the detailed information in the “Abstract” section, maybe the summarizing information is ok.

7 Lines 29-31: Are you sure that antennae sensilla is related with feeding behavior and ability to carry and transmit Ca Liberibacter? Or is related with volatile perception? Maybe feeding behavior and bacteria acquisition and transmission is related with feeding behavior, interaction with gut and other microorganisms, please justify the relation.

Response: We are very sorry for putting forward the argumentative and uncertain conclusion. Thank you very much for your comments, and we have revised the expression.

8 Line 35, add “also”.

Response: Done. Thank you very much.

9 Use the complete name (Diaphorina citri) when start a phase.

Response: Done. Thank you very much.

10 Line 43: associated with (note: there is no Koch’s postulate for these bacteria complex).

Response: Done. Thank you very much.

11 Lines 47-48: Please, there is a controversy for this subject. Please clarify.

Response: Done. Thank you very much for your comment. We have deleted the sentence.

12 Lines 49-50: You should link gustatory function of antennae with feeding behavior and infective insects in order to justify your theory. I am not convict about the relationship of antennae sensilla and HLB transmission. And why you choose nymphs instead adults (or both), since adults are responsible for.

Response: Thank you very much for your comment. We agree with you, there isn’t an exactly relationship of antennae sensilla and HLB transmission. Actually, we also finished the study of adult D. citri, but they were not written in our MS because of the similar research reporting by Onagbola et al. (2008).

(Onagbola EO, Meyer WL, Boina DR, Stelinski LL. Morphological characterization of the antennal sensilla of the Asian citrus psyllid, Diaphorina citri Kuwayama (Hemiptera: Psyllidae), with reference to their probable functions. Micron. 2008; 39(8): 1184-1191.)

13 Line 66: Perfect. But I am not convicted about the explanation used to justify the nymphs studies. The absence of studies is ok, but I do not agree with ability of transmission vs feeding behavior vs antenna sensilla in nymphs. The introduction needs to be completely written in order to justify the above comment.

Response: Thank you very much, and we have revised the relevant content according to the comment.

14 Line 80: What is the rearing procedure adopted (cite authors). How many generations under laboratory conditions before the experiments? Please clarify the criteria adopted for instar characterization (figure of each instar) and morphological aspects (body length and width, wing buds aspects…days after egg hatch for each nymphal stage…

Response: Thank you very much for your comment. We are very sorry that we don’t record the ACP generations after rearing under laboratory conditions before the experiments. Rearing under laboratory conditions, it usually takes about 15-20 days for ACP to finish one generation. So, it may be 14-18 generations when we begin to take the experiments in March 2017. We refer Ruan et al. (2012) to clarify criteria of ACP instar characterization. Besides the difference of body size between different stages of nymphs, the most important instar characterization is the difference of wing buds. There is no wing buds in the first-instar ACP nymphs. The wing buds are found on the second segment of the thorax in the second-instar ACP nymphs. The wing buds in the third-instar nymphs increase significantly. Their leading edges extend to the posterior edges of their compound eyes, while their posterior edges extend to the 3rd segment of the abdomen. Moreover, their antennal tips turn black and appear caudate beard. In fourth-instar ACP nymphs, the leading edges of wing buds exceed their compound eyes, the posterior edges exceed the 3rd segment of the abdomen. The fifth-instar ACP nymphs are significantly bigger than the other stage nymphs. Their antennae are black except the scape, and the TH1 and TH2 can be clearly observed.

Figure 1 The egg and five different nymphal stages of Diaphorina citri under 50 × magnification. (a) Egg; (b) First-instar nymphs; (c) Second-instar nymphs; (d) Third-instar nymphs; (e) Fourth-instar nymphs; (f) Fifth-instar nymphs.

(Ruan CQ, Chen JL, Liu B, Duan YP, Xia YL. 2012. Morphology and behavior of Asian citrus psyllid, Diaphorina citri Kuwayama. Chinese Agricultural Science Bulletin, 28(31): 186-190.)

15 number of replications.

Response: Done. Thank you very much.

16 Why parametrical and non-parametrical testes were adopted for different parameters? Please clarify the criteria (data normality…).

Response: Thank you very much. For the statistical analysis, we consulted an expert Prof. Wei-Jian Wu from South China Agricultural University who are also major in Biostatistics. The lengths and widths are conformed to normal distribution, the non-parametrical testes is ok.

17 Line 105: Seems obvious: a pair of antennae is a characteristic of Insecta. This sentence is not helpful.

Response: Thank you very much. We have revised the sentence, you can find in lines 99-100.

18 Lines 221-230: the authors replicated information presented in topic Results.

Response: Thank you very much for your comment. The first paragraph of “Discussion” section, it is the generalization and summarization of the whole results in our study. It may be some replicated information presented in “Results” section.

19 Line 231: This information could be clarified if authors decided to include adults? SB and CvS have degenerated our Onagbola did not see these structures? Can you support the degeneration theory?

Response: Thank you very much for your comment. We found SBs and CvSs in ACP nymphs but not in adults; they were not reported by Onagbola et al. (2008) also. We have added the general analysis of adult antennae and antennal sensilla in S1 File.

20 Line 232: same comment. The authors are assuming the fact, but the fact is not proved. I suggest the addition of study with adults in order to conclude the degeneration/replacement by antennal rhinaria.

Response: Thank you very much for your comment. We add the main results of adult ACP in S1 File.

21 Lines 235-237: Provide (Order: Family) for the listed species.

Response: Done. Thank you very much.

22 Line 289: Point 1: However, there is an interaction between the bacteria, nutritional medium (presented in insect gut) and the gut that can explain the ability of third nymphs to adults to carry and transimit the bacteria. Point 2: there is a minimum period (feeding period for the nymph to acquire the bacteria. Point 3: there is a latent period of the bacteria before multiplication. Point 4: Thus, that is a minimum concentration of Ca Lbieribacter bacterial title to be detected by Qpcr. Point 5: first to second instar nymphs feed lower than higher nymphal stages and adults, ingesting lower concentration of bacteria (+latent period) + minimum concentration of bacterial title is necessary for qPCR detection could explain this phenomena. I am not convicted about the relationship between sensilla number/types with HLB aquistition/transition by older nymphal stages.

Response: Thank you very much for your comments. They are very valuable and helpful for the important guiding significance to improve our MS and to our future researches. We agree with your opinion, many works must be done to convict about the relationship between sensilla number/types with HLB aquistition/transition by older nymphal stages. We have revised the “Introduction” and “Discussion” section for the expression about the deduction combined with the literatures and the reviewers’ comments.

23 The figure should be self-explained. Add insect species and stage (instar).

Response: Done. Thank you very much.

24 Diaphorina citri instead D. citri.

Response: Done. Thank you very much.

25 Line 526: Diaphorina citri instead D. Scientific tables: please use only vertical.

Response: Done. Thank you very much. The layout of tables is used according to the requirement in PLoS ONE. The horizontal line can be used in table.

26 Figures: Diaporina citri. Inform instar that you used to take the photos.

Response: Done. Thank you very much.

Special thanks to you for your comments and suggestions.

Responds to the “Reviewer general comments” file:

1 Title: Asian citrus psyllid family: Liviidae instead Psyllidae

Response: Done, thank you very much.

2 The authors support the study of morphological characteristics of D. citri nymphs based on the fact that immature stages have the ability to transmit Ca. Liberibcter more efficiently than adults. This argument is used to justify the choice for immature stages. However, we cannot forget that adults can spread bacteria more efficiently due its flight (from plant to plant). Another aspect that supports the HLB spread is the fact that D. citri adults, at first, prefers HLB plants (regulated by MeSa Vocs) and after feeding on HLB plants, move the health plants (better nutritional aspects). I am not sure about the author´s sentence to justify the studies only with nymphs.

Response: Thank you very much for your good suggestion, we are extremely grateful to Reviewers for pointing out this problem. As the Reviewer’ comments said that the argument used to lead to the thesis is not very reasonable. We have re-written the relevant contents.

3 The entire literature related to ACP/chemical ecology are based on adult ACP behavior challenged by chemicals from plants or from co-specific insects and their VOCs. Some manuscripts explain the role of ACP adult antenna receptors challenged by plant and/or chemical volatiles. It is not possible to understand the relationship of immature ACP stages with chemicals, since nymphs do not perform choices in response to chemicals, since nymphs do not perform movements. Thus, I still not convicted about the choice for nymphs only. However, the comparison of nymph antenna structures/sensillas with adults could be helpful for other studies and if done, could be more easy to justify the study. 

Response: Thank you very much for your comments. We have performed the studies of adult D. citri with SEM and TEM, but we do not write in our MS. It may have many repeated contents with the article of Onagbola et al. (2008), if we reported the structures of antennae and antennal sensilla of adult ACP. The ACP has five nymphal stages, early stages are docile and move only when disturbed or overcrowded (Tsai & Liu, 2000), whereas older nymphs and adults are more mobile (Hall et al., 2013).

(Onagbola EO, Meyer WL, Boina DR, Stelinski LL. Morphological characterization of the antennal sensilla of the Asian citrus psyllid, Diaphorina citri Kuwayama (Hemiptera: Psyllidae)

(Tsai JH, Liu YH. 2000. Biology of Diaphorina citri (Hemiptera: Psyllidae) on four host plants. Journal of Economic Entomology, 93: 1721-1725.)

(Hall DG, Richardson ML, Ammar ED, Halbert SE. 2013. Asian citrus psyllid, Diaphorina citri, vector of citrus huanglongbing disease. Entomologia Experimentalis et Applicata, 146(12): 207-223.)

4 However, the authors cited 2 articles that assessed the antennal structures of adults. However, due to the lack of information (see your discussion topic), the authors are unable to explain the absence of these structures during the investigation of these articles. Maybe there is the absence of these structures in these articles, or the cited authors did not see the mentioned structures. In order to solve this doubt, I encourage the authors to add morphological aspects of adult antenna, since the literature was not helpful to support the findings with nymphs and comparison with adults. 

Response: Thank you very much for your suggestions. We have performed the studies of adult D. citri with SEM and TEM, and add part of results as the supplementary data (see S1 File and S1 Table).

5 In general, the introduction is not justified neither well presented. I suggest that this topic could be re-written and that in M&M and Results could be added adult antennal structures. 

Response: Thank you very much, and we have re-written this part according to your suggestion. For the adult antennal structures, Onagbola et al. (2008) had reported the research by SEM and TEM, so we think it does not need to write the similar results, just as the supplementary data in S1 File and S1 Table.

(Onagbola EO, Meyer WL, Boina DR, Stelinski LL. Morphological characterization of the antennal sensilla of the Asian citrus psyllid, Diaphorina citri Kuwayama (Hemiptera: Psyllidae)

6 There is a lack of substantial information that allow the replication of the protocols. There is no enough information related to the insect (rearing protocol, age, description of what the authors considered as nymph 1, nymph 2, nymph 3, nymph 4 and nymph 5 (see pdf manuscript with my notes).

Response: We are very sorry for our negligence of lacking the important information. We have added the relevant contents in lines 90-92. The classification characteristics of different stages of ACP nymphs are according to Ruan et al. (2012).

 (Ruan CQ, Chen JL, Liu B, Duan YP, Xia YL. 2012. Morphology and behavior of Asian citrus psyllid, Diaphorina citri Kuwayama. Chinese Agricultural Science Bulletin, 28(31): 186-190). 

7 Number of replicates.

Response: Done, thank you very much, and we have added the relevant content in lines 90-92.

8 More details about statistical analyses (see pdf manuscript with my notes). 

Response: Done, thank you very much for your comments.

9 The beginning of discussion presents text that is related to results (see pdf manuscript with my notes). The authors are not convicted if antennal structures present in nymphs were degenerated or Onagbola et al did not see this structure in adults (but could be present). This fact did not allow the authors to be convincing during the discussion of the topic. A solution could be the assessment of adult antennal morphological aspects in order to be clear and consistent/convicted to explain the ‘possible degeneration’ of these structures in ACP adults.

Response: For the adult and nymphal ACP antennal structures, we have performed at least five times with SEM, three times with TEM. The results were the same, and we added some main adult results as the supplementary data in S1 File and S1 Table.

10 Figures should be self-explained (comments in pdf manuscript) 

Response: Done, thank you very much.

We tried our best to improve the manuscript and made some changes in the manuscript. We appreciate for Editors/Reviewers’ warm work earnestly, and hope that the correction will meet with approval.

Once again, thank you very much for your comments and suggestions.

---

## [Decision Letter · Decision Letter 2]

6 Feb 2020

PONE-D-19-23769R2

Morphological characterization of nymphal antennae and antennal sensilla of Diaphorina citri Kuwayama (Hemiptera: Liviidae)

PLOS ONE

Dear Dr. Zheng,

Thank you for submitting your manuscript to PLOS ONE. None of the reviewers recommended publication of your manuscript mostly because you did not address all of their concerns.  Under these circumstances I would normally recommend rejection, but I would like to invite you to submit a revised version of the manuscript that addresses the points raised during the review process.

Please note that I will not make a decision without consulting with the same reviewers and if you do not address their concerns properly, they may not change their opinion and, consequently, I cannot recommend publication. 

We would appreciate receiving your revised manuscript by Mar 22 2020 11:59PM. To enhance the reproducibility of your results, we recommend that if applicable you deposit your laboratory protocols in protocols.io, where a protocol can be assigned its own identifier (DOI) such that it can be cited independently in the future. For instructions see: http://journals.plos.org/plosone/s/submission-guidelines#loc-laboratory-protocols

We look forward to receiving your revised manuscript.

Kind regards,

Walter S. Leal

Academic Editor

PLOS ONE

P.S.: On a personal note, we are following with sadness the corona epidemics and praying that the situation will be under control the soonest and that the Chinese people will be back to their normal life. If this situation affects your ability to respond within the suggested deadline of March 21st, please do not hesitate to request an extension. 

Reviewers' comments:

Reviewer's Responses to Questions

**Comments to the Author**

1. If the authors have adequately addressed your comments raised in a previous round of review and you feel that this manuscript is now acceptable for publication, you may indicate that here to bypass the “Comments to the Author” section, enter your conflict of interest statement in the “Confidential to Editor” section, and submit your "Accept" recommendation.

Reviewer #1: All comments have been addressed

Reviewer #2: (No Response)

Reviewer #3: (No Response)

2. Is the manuscript technically sound, and do the data support the conclusions?

Reviewer #1: Partly

Reviewer #2: Partly

Reviewer #3: Yes

3. Has the statistical analysis been performed appropriately and rigorously? 

Reviewer #1: Yes

Reviewer #2: Yes

Reviewer #3: Yes

4. Have the authors made all data underlying the findings in their manuscript fully available?

Reviewer #1: Yes

Reviewer #2: Yes

Reviewer #3: Yes

5. Is the manuscript presented in an intelligible fashion and written in standard English?

Reviewer #1: Yes

Reviewer #2: No

Reviewer #3: Yes

6. Review Comments to the Author

Reviewer #1: The manuscript has been greatly improved, and the researcher can benefit from this interesting research on the antennae ultrastructure of ACP. However, there is no more figures to show the pores which are hypothesized where the semiochemicals can enter the antennae. It will be difficult to support the claim that some sensilla are involved (or function)in “olfactory behavior and host identification”. I strongly suggested that the authors should add more figures to show the pores on the sensilla surface with SEM or TEM. There is another paper on Micron which showed that they observed the multiparous structure with the same pest https://www.sciencedirect.com/science/article/pii/S0968432808001182. Thus, I suggest the authors should make a more interesting story to publish their findings.

The full journal name should be shown in Reference 3.

Reviewer #2: Comment #1: I am not convinced by the authors’ justification for not wanting to revise the results section. As it stands, the very first thing that the readers come across in the results section is the name (or some abbreviations) of sensilla (TH1, TH2, ST, etc). Instead, readers should first be introduced to these sensilla and what their characteristics are FOLLOWED BY information regarding their presence/absence.

Comment #2: As I mentioned before, I am fully aware of the constraints the authors might have to performing TEM and or electrophysiology and that was why I suggested it to be submitted to, e.g., Micron as it publishes findings solely on SEM. The previous works the authors referenced, such as the one published in Micron, have a combination of SEM and TEM. Likewise, the works published in other journals include molecular work or amino acid alignment and not only SEM.

Reviewer #3: The manuscript entitled “Morphological characterization of nymphal antennae and antennal sensilla of Diaphorina citri Kuwayama (Hemiptera: Liviidae)’ reports general aspects of antenna/sensilla morphology of Diaphorina citri. The revision 2 was significantly improved, however major points need to be carefully corrected before its publication in PlosOne.

In my opinion, a final revision round is desirable in order to be sure that the paper is clear to be published.

7. PLOS authors have the option to publish the peer review history of their article (what does this mean?). If published, this will include your full peer review and any attached files.

Reviewer #1: Yes: Zhao Liu

Reviewer #2: No

Reviewer #3: Yes: Volpe, H. X. L.

---

## [Author Response · Author response to Decision Letter 2]

12 Mar 2020

We very much appreciate the careful reading of our MS and valuable suggestions of the reviewers. We have carefully considered the comments and have revised the MS accordingly. The main corrections in the paper and the responds to the comments are as following:

Responds to Reviewer 1 (from the decision letter)

1. The manuscript has been greatly improved, and the researcher can benefit from this interesting research on the antennae ultrastructure of ACP. However, there is no more figures to show the pores which are hypothesized where the semiochemicals can enter the antennae. It will be difficult to support the claim that some sensilla are involved (or function) in “olfactory behavior and host identification”. I strongly suggested that the authors should add more figures to show the pores on the sensilla surface with SEM or TEM. There is another paper on Micron which showed that they observed the multiparous structure with the same pest https://www.sciencedirect.com/science/article/pii/S0968432808001182. Thus, I suggest the authors should make a more interesting story to publish their findings.

Response: Thank you very much for your comments. We also know that it will well improve the quality of our MS using SEM and TEM together. We have tried our best to get some high quality TEM photographs. Unfortunately, we failed. We performed the TEM studies at least three times at three different electron microscopy centers including Centre Laboratory of the South China Botanical Garden where we successfully got the good quality TEM photographs in 2015. Our teams have done a lot of studies about SEM (Zheng et al., 2014; 2016; Xue et al., 2015; Hang et al., 2018) and TEM (Zheng et al., 2016), but only one time to get the perfect TEM photographs (Zheng et al., 2016). Because of the outbreak of the novel coronavirus in our country, we can’t perform the TEM study within these days. The other reason we don’t want to try it again is because its cost is too expensive to perform these studies in the electron microscopy centers. The costs of SEM and TEM are often more than 10000 CNY (about 1430 $) for one time. The total costs of SEM and TEM (three times together) was close to 35000 CNY (about 5000 $). We are very sorry that we decide not to try it again for this situation. Thank you very much for your kind suggestion, it is very helpful and valuable for our future researches.

Han-Ying Yang#, Li-Xia Zheng#, Zhen-Fei Zhang, Yang Zhang, Wei-Jian Wu. The structure and morphologic changes of antennae of Cyrtorhinus lividipennis (Hemiptera: Miridae: Orthotylinae) in different instars. PLoS ONE, 2018,13(11): e0207551.

Xue H, Zheng LX, Wu WJ. Morphometry of compound eyes of three Bactrocera (Diptera: Tephritidae) species. Florida Entomologist, 2015, 98(2): 806-809.

Zheng LX, Wu WJ, Liang GW, Fu YG. Nymphal antennae and antennal sensilla in Aleurodicus dispersus (Hemiptera: Aleyrodidae). Bulletin of Entomological Research, 2014, 104: 622-630.

Zheng YH#, Zheng LX#, Liao YL, Wu WJ. Sexual dimorphism in antennal morphology and sensilla ultrastructure of a pupal endoparasitoid Tetrastichus howardi (Olliff) (Hymenoptera: Eulophidae). Microscopy Research and Technique, 2016, 79: 374-384.

2. The full journal name should be shown in Reference 3.

Response: Done. Thank you very much.

Responds to Reviewer 1 from the file (Plos one Review 26-01-2020):

1. The manuscript has been greatly improved, and researcher can benefit from this research on the antennae ultrastructure of ACP. However, there is no more figures to show the pores which are hypothesized where the semiochemicals can enter the antennae. It will be difficult to support the claim that some sensilla are involved (or function) in “olfactory behavior and host identification”. I strong suggested that the authors should add more figures to show the pores on the sensilla surface with SEM or TEM. There is another paper on Micron which showed that they observed the multiparous structure with the same pest https://www.sciencedirect.com/science/article/pii/S0968432808001182. Thus I suggest the authors should make a more interesting story to publish their findings.

Response: Thank you very much for your suggestion. Yes, this reference is also very valuable and helpful to our MS. Our SEM photographs of adult ACP (see S1 File) are also perfect quality, but we don’t get the good quality TEM photographs both nymphal and adult antennal sensilla. We are very sorry that we could not provide any more perfect photographs of SEM or TEM. Thank you very much again for your suggestion, it is very nice and valuable. Here, we hope our photos may meet the requirement of PLoS One.

Special thanks to you for your comments and suggestions.

Responds to Reviewer 2 (from the decision letter)

1. I am not convinced by the authors’ justification for not wanting to revise the results section. As it stands, the very first thing that the readers come across in the results section is the name (or some abbreviations) of sensilla (TH1, TH2, ST, etc). Instead, readers should first be introduced to these sensilla and what their characteristics are FOLLOWED BY information regarding their presence/absence.

Response: Done. Thank you very much for your comments again. Yes, as you see, it is much concise in the results section after revising according to your suggestion.

2. As I mentioned before, I am fully aware of the constraints the authors might have to performing TEM and or electrophysiology and that was why I suggested it to be submitted to, e.g., Micron as it publishes findings solely on SEM. The previous works the authors referenced, such as the one published in Micron, have a combination of SEM and TEM. Likewise, the works published in other journals include molecular work or amino acid alignment and not only SEM.

Response: Thank you very much for your suggestions again. We also agree with you. It will be really well to improve our MS adding TEM and electrophysiology studies. The reason of why we don’t add the TEM study is stated above. Also, we could not add the nymphal electrophysiological study due to the difficulty in connecting the nymphal antennae. We are very sorry, but your suggestions and comments are very valuable and helpful for our future researches.

Special thanks to you for your comments and suggestions.

Responds to Reviewer 3 (from the decision letter)

1. The manuscript entitled “Morphological characterization of nymphal antennae and antennal sensilla of Diaphorina citri Kuwayama (Hemiptera: Liviidae)’ reports general aspects of antenna/sensilla morphology of Diaphorina citri. The revision 2 was significantly improved, however major points need to be carefully corrected before its publication in Plos One.

In my opinion, a final revision round is desirable in order to be sure that the paper is clear to be published.

Response: Thank you very much for your kind comments and suggestions. We learned it carefully and done one to one response to the comments.

Responds to Reviewer 3 (file “PONE-D-23769_R2”):

1. Line 43: It is plural, since is associated to 3 bacteria.

Response: Done. Thank you very much.

2. Line 111: these sensilla refers to table 2? Cite it to clarify information.

Response: Thank you very much. According to the other reviewer’s suggestion, we rewrote part of the results and deleted the sentence.

3. Line 117: 35.96 longer: These information was not presented in Table 1. Suggestion: Include a new column in this table adding the antenna growth (+SE) for each instar (2-5 instar) (instar 2 size - instar 1 size = ?) 

Response: Done. Thank you very much for your kind suggestion.

4. Line 119: I found this information in Table 2. Please cite it.

Response: Done. Thank you very much. According to the other reviewer’s suggestion, we rewrote part of the results and deleted the sentence.

5. Line 125: 1.5 times longer. Suggestion: Standardization, xxx um longer (as expressed for topic (second instar). Use the information that you will provide in the new column (Table 1) (instar 3 size - instar 2 size = ?) antenna growth. Dont forget to include SE in table and text.

Response: Done. Thank you very much.

6. Line 127: Are you referring to Table 2? Cite it here in order to clarify the information.

Response: Done. Thank you very much. According to the other reviewer’s suggestion, we rewrote part of the results and deleted the sentence.

7. Line 132: Information missed: The authors presents the antenna growth for instar 2 and 3. But what is the growth in 4 instar (4 instar size - 3 instar size = growth)

Response: Done. Thank you very much. 

8. Line 139: For standardization: add antenna growth in this topic. (instar 5 - instar 4 = ?

Response: Done. Thank you very much.

9. Line 141: For standardization: add antenna growth in this topic. (instar 5- instar 4 = ?).

Response: Done. Thank you very much for the reviews of 2-9, we have rewritten this part of results according to the other reviewer’s suggestion. Also, we have added some information according to your suggestions of 2-9.

10. Line 140: Please check your Figure. I can’t find Fig 1 F. Figure legend is wrong (A-C, ok) (D-H, wrong). I think (D-E).

Response: Yes, you are right. Thank you very much.

11. Line 156: Fig 2 A refers to a 3rd instar nymph, but you assume in the sentence that is related to 5th instar. In this way, I added a text as a solution to solve this issue.

Response: Done. Thank you very much for your kind suggestion.

12. Line 167: Check my suggestion, since: Table 2 supports that you found ST for all nymphal stages! And Fig 3 is related only for a third instar nymph.

Response: Thank you very much for your kind suggestion, but I think that Table 2 just shows the abundance and distribution of ST, here we showing morphology of ST in this sentence, Table 2 may be cited in the last sentence. What do you think?

13. Line 174: Same issue here: Table 2 indicates that you found PSO for all instars and Fig 4 refers to fifth instar nymph.

Response: Thank you very much for your suggestions. The reviews of 10-13 may be the same issue. It is not so good to present all photographs of all types of antennal sensilla of each nymphal stage of D. citri. For the same type of antennal sensilla, one of the stages of nymphal D. citri was chosen to perform the morphology and structure in figure, even if they can be found in each nymphal stage. The SEM quality of antennal sensilla in the first- and second-instar nymphs is a little worse than the others, so they were not presented in figures. The photographs we chose should be the good quality, and we think it is best to include each nymphal stage of D. citri.

14. Lines 226-231: Highlighted in yellow and red: Results replicated.

Response: Thank you very much, and we have deleted.

15. Line 286: First citation: add (Order: Family).

Response: Thank you very much. It has been cited in lines 240-241 in Reference 32, not first citation.

16. Table 1: Add a column Growth um and add analyses.

Response: Done. Thank you very much for your kind suggestion, it is also valuable and helpful for our MS and other researches.

17. Tables: use only horizontal lines…Since it is a scientific table.

Response: Thank you very much for your suggestion. I agree with you, but it is the format requirement of the table in PLoS ONE. In addition, we unload the usual scientific table in the file of “Tables”. If they need, they can also choose to use.

18. Table 2: format this colmn. (SB3).

Response: Done. Thank you very much.

Responds to Reviewer 3 (file “Reviewer comments R2”):

Response: Thank you very much for your kind and valuable suggestions. I read the comments of file “Reviewer comments R2” carefully and find that they are the replicated suggestions and comments with the file “PONE-D-23769_R2”. So, you can find the responds above. Special thanks to you for your comments and suggestions.

We tried our best to improve the manuscript and made some changes in the manuscript. We appreciate for Editors/Reviewers’ warm work earnestly, and hope that the correction will meet with approval.

Once again, thank you very much for your comments and suggestions.

---

## [Decision Letter · Decision Letter 3]

28 Apr 2020

PONE-D-19-23769R3

Morphological characterization of antennae and antennal sensilla of Diaphorina citri Kuwayama (Hemiptera: Liviidae) nymphs

PLOS ONE

Dear Dr. Zheng,

Thank you for submitting your manuscript to PLOS ONE. After careful consideration, we feel that it has merit but does not fully meet PLOS ONE’s publication criteria as it currently stands. Therefore, we invite you to submit a revised version of the manuscript that addresses the points raised during the review process.

Please address all issues raised by the two reviewers and those that I added at the bottom of this email. 

We would appreciate receiving your revised manuscript by Jun 12 2020 11:59PM. To enhance the reproducibility of your results, we recommend that if applicable you deposit your laboratory protocols in protocols.io, where a protocol can be assigned its own identifier (DOI) such that it can be cited independently in the future. For instructions see: http://journals.plos.org/plosone/s/submission-guidelines#loc-laboratory-protocols

We look forward to receiving your revised manuscript.

Kind regards,

Walter S. Leal

Academic Editor

PLOS ONE

Reviewers' comments:

Reviewer's Responses to Questions

**Comments to the Author**

1. If the authors have adequately addressed your comments raised in a previous round of review and you feel that this manuscript is now acceptable for publication, you may indicate that here to bypass the “Comments to the Author” section, enter your conflict of interest statement in the “Confidential to Editor” section, and submit your "Accept" recommendation.

Reviewer #1: All comments have been addressed

Reviewer #3: (No Response)

2. Is the manuscript technically sound, and do the data support the conclusions?

Reviewer #1: Yes

Reviewer #3: Yes

3. Has the statistical analysis been performed appropriately and rigorously? 

Reviewer #1: Yes

Reviewer #3: Yes

4. Have the authors made all data underlying the findings in their manuscript fully available?

Reviewer #1: Yes

Reviewer #3: Yes

5. Is the manuscript presented in an intelligible fashion and written in standard English?

Reviewer #1: Yes

Reviewer #3: No

6. Review Comments to the Author

Reviewer #1: The manuscript has been improved greatly both in writing and figure quality, and I think is suitable to be accepted. However, some more detailed information should be clarified.

1. The sensilla trichoidea 1, 2,... It means the subtype or just different sensilla on different location?

2. The sensila basiconica usually refers to the sensilla with the "cone" shape. The Fig. 3 and Fig.6 A is very similar. Why one is sensill basiconica and one is sensilla trichodiae? Based on reference 19, the SB in Fig.6 A named sensilla trichoidea will be suitalbe.

3 How to minumum the difference caused by the angle difference when the (nymph)antenne were mounted on to the holder? It is too diffcult to keep the nymph on the double stape with posture (to keep the sensilla to be scaned with same view) under a optical microscope. The length of the sensilla will be greatly different if scaned with different view.

4. With the Nova 85 Nano 430, it is posiable to find the pores on the sesilla trichoidae without gold coat. The sensilla on nymph antennae are very differnt with the sensilla on the adult antennae. However, in Ref. 19, they used the adult antennae; The antenne are from the nymph. I think the reader can get more useful information if the results about the sensilla from both nymph and adult are presented here.

In Line 49, "For most insects, the antennae are peripheral sensory." It is not suitalbe. In the paper Zoomorphology. 2017; 136(3): 327–347., peripheral refers the senilla on the antennae not the antennae.

In Line 55, the "spawning" usually used when discribe the aquatic animal. Rewrrtie this sentence please.

In line 226, "the number of the antennal sensiall was small" Compared with which insect or the insect?

The figure lengend shuold be polised again to help the reader obtain the information quikly(refer to referece paer 19 and 22). Figure legend is needed in Fig 3. ,Fig.4, Fig.7. There are only figure title with these fugures.

Reviewer #3: (No Response)

7. PLOS authors have the option to publish the peer review history of their article (what does this mean?). If published, this will include your full peer review and any attached files.

Reviewer #1: No

Reviewer #3: Yes: Haroldo Xavier Linhares Volpe

PONE-D-19-23769_R3

MAKE SURE TO HAVE ALL SCIENTIFIC NAMES IN ITALIC

The antennae of D. citri nymphs were observed when individuals were fixed in a ventral position and looking the region located between the compound eyes.

The antennae of D. citri nymphs were observed when individuals were fixed in a ventral position and looking **at** the region located between the compound eyes.

In addition, the length of the longest antennae (the fifth-instar nymphal antennae) was five times more than the shortest antennae (the first-instar nymphal antennae).

Additionally, the fifth-instar nymphal antennae were five times longer than the first-instar nymphal antennae.

We also found the total number of the antennal sensilla was increased with the increase of the nymphal instar.

We also found the total number of the antennal sensilla increased from the first- to the fifth-instar nymphs.

We do not find the SB1-4, CvS1 and CvS2 in adult D. citri (S1 File 202 and S1 Table), as well as Onagbola et al. [20], maybe they were degenerated and missed in adults.

Like Onagbola et al. [20], we did not find the SB1-4, CvS1 and CvS2 in adult D. citri (S1 File 202 and S1 Table). Perhaps, they were degenerated and missed in adults.

In addition, considering the TH1 and TH2 in adults (Fig 2 in S1 File) that revealed wall pores suggesting a plausible role in perception of olfactory stimuli.

Considering that TH1 and TH2 (Fig 2 in S1 File) have wall pores in adult antennae, it is conceivable that they are olfactory sensilla.

The PSO was first reported in P. pyricola and suggested chemoreceptors [27].

The PSO was first reported in P. pyricola and suggested to be chemoreceptors [27].

The SBs only occurred on the antennae of third-, fourth- and fifth-instar nymphs. Similarly, they were not reported in other psyllids, no matter nymphs or adults  [20,27,31].

The SBs only occurred on the antennae of third-, fourth- and fifth-instar nymphs. Similarly, they were not reported in other nymphs or adults psyllids [20,27,31].

The SCA has been found in many insects, including whiteflies [36] and psyllids..

The SCA has been observed in many insects, including whiteflies [36] and psyllids…

The SCA was seemed to occur in

The SCA seems to occur in

whereas the SCA with pores was involved in gustatory function were highly susceptible to humidity

whereas the SCA with pores was involved in the gustatory system as well as in humidity reception

are related to the feeding behavior on the citrus with Las by these nymphs

are indeed related to feeding behavior of nymphs on Las-infected citrus

psyllid's antennae and antennal sensilla and allow us to better understand the host location mechanism in D. citri.

psyllid's antennae and antennal sensilla and provides new insights on how D. citri nymphs might locate host plants.

Table 2. Abundance and distribution of sensilla on the antennae for the five nymphal stages of Diaphorina citri

Table 2. Abundance and distribution of sensilla on the antennae of the fifth nymphal stages of Diaphorina citri

(SAME FOR TABLE 3)

---

## [Author Response · Author response to Decision Letter 3]

15 May 2020

We very much appreciate the careful reading of our MS and valuable suggestions of the reviewers and editor. We have carefully considered the comments and have revised the MS accordingly. The main corrections in the paper and the responds to the comments are as following:

Responds to Reviewer 1 (from the decision letter and file “review”)

1. The sensilla trichoidea 1,2,… It means the subtype or just different sensilla on different location?

Response: Thank you very much, but we just reported one type of sensilla trichoidea in our MS. Did you mean the sensilla basiconica 1-3? For the SB1-3, they have a big inflexible socket but different surfaces of the stem and locations. So they should be identified as different subtype not different sensilla judging by the morphological surface features.

2. The sensilla basiconia usually refers to the sensilla with the “cone” shape. The Fig.3 and Fig.6A is very similar. Why one is sensilla basiconica and one is sensilla trichodiae? Based on the refernce 20, the SB in Fig.6A named sensilla trichoidea will be suitable.

Response: Thank you very much for your good suggestion. Yes, as you suggested that the SB in Fig 6A (SB1) is similar with the ST in Fig 3, and most of the ST in adult ACP in the reference 20 were distributed in the scape and pedicel. It may mean that the ST in the adults is developed from the ST in the first segment of nymphal antennae. We have corrected the relative information, and thanks again.

3. How to minimum the difference caused by the angle difference when the (nymph) antennae were mounted on the holder? It is too difficult to keep the nymph on the double stape with posture (to keep the sensilla to be scanned with the same view) under an optical microscope. The length of the sensilla will be greatly different if scanned with different view.

Response: Thank you very much. Actually, we can fix the antennae just only two sides (the ventral and dorsal position). There are few antennal sensilla in ACP nymphs, for the same type of the sensilla, the data of the length or width of the sensilla is measured from the one distributed on the ventral and dorsal positions each half. Maybe it can minimum the difference caused by the angle difference in this way.

4. With the Nova 85 Nano 430, it is possible to find the pores on the sesilla trichoidae without gold coat. The sensilla on nymph antennae are very different with the sensilla on the adult antennae. However, in Ref. 20, they used the adult antennae; The antennae are from the nymph. I think the reader can get more useful information if the results about the sensilla from both nymph and adult are presented here.

Response: Thank you very much for your suggestion. Most of our results about the adults are similar to those in Ref. 20. (Eight morphologically distinct types of sensilla were found to distribute on the antennae of female and male D. citri in our study, with seven types reminiscent of those described by Onagbola et al.). In this case, we add our adult results as the supplemental information (S1 File) submitting together.

5. In Line 49, “For most insects, the antennae are peripheral sensory.” It is not suitable. In the paper Zoomorphology. 2017; 136(3): 327-347. Peripheral refers the sensilla on the sensilla on the antennae not the antennae.

Response: Done. Thank you very much. We have corrected the sentence according to the comments.

6. In Line 55, the “spawning” usually used when described the aquatic animal. Rewrite this sentence please.

Response: Done. Thank you very much.

7. In line 226, “the number of the antennal sensilla was small “compared with which insect or the insect?

Response: Thank you very much. I am really sorry that I can’t find the sentence you refer. Actually, the number of each type of the antennal sensilla and the total number in ACP nymphs are small (Table 2).

8. The figure legend should be polised again to help the reader obtain the information quickly (refer to referece paer 19 and 22). Figure legend is needed in Fig.3, Fig.4, Fig.7. There are only figure title with these figures.

Response: Done. Thank you very much.

Special thanks to you for your comments and suggestions.

Responds to “PONE-D-19-23769_R3” (from the decision letter)

1. make sure to have all scientific names in italic.

Response: Done. Thank you very much.

2. Line 99-100: “The antennae of D. citri nymphs were observed when individuals were fixed in a ventral position and looking the region located between the compound eyes.” Should be “The antennae of D. citri nymphs were observed when individuals were fixed in a ventral position and looking at the region located between the compound eyes.

Response: Done. Thank you very much.

3. Line 109-111: “In addition, the length of the longest antennae (the fifth-instar nymphal antennae) was five times more than the shortest antennae (the first-instar nymphal antennae).” Should be “Additionally, the fifth-instar nymphal antennae were five times longer than the first-instar nymphal antennae.”

Response: Done. Thank you very much.

4. Line 124-125: “We also found the total number of the antennal sensilla was increased with the increase of the nymphal instar.” Should be “We also found the total number of the antennal sensilla increased from the first- to the fifth-instar nymphs.”

Response: Done. Thank you very much.

5. Line 119: I found this information in Table 2. Please cite it.

Response: Done. Thank you very much.

6. Line 201-203: “We do not find the SB1-4, CvS1 and CvS2 in adult D. citri (S1 File and S1 Table), as well as Onagbola et al. [20], maybe they were degenerated and missed in adults.” Should be “Like Onagbola et al. [20], we did not find the SB1-4, CvS1 and CvS2 in adult D. citri (S1 File and S1 Table). Perhaps, they were degenerated and missed in adults.

Response: Done. Thank you very much.

7. Lines 217-219: “In addition, considering the TH1 and TH2 in adults (Fig 2 in S1 File) that revealed wall pores suggesting a plausible role in perception of olfactory stimuli.” Should be “Considering that TH1 and TH2 (Fig 2 in S1 File) have wall pores in adult antennae, it is conceivable that they are olfactory sensilla”

Response: Done. Thank you very much.

8. Line 226: “The PSO was first reported in P. pyricola and suggested chemoreceptors [27].” Should be “The PSO was first reported in P. pyricola and suggested to be chemoreceptors [27].

Response: Done. Thank you very much. 

9. Lines 244-246: “The SBs only occurred on the antennae of third-, fourth- and fifth-instar nymphs. Similarly, they were not reported in other psyllids, no matter nymphs or adults [20,27,31].” Should be “The SBs only occurred on the antennae of third-, fourth- and fifth-instar nymphs. Similarly, they were not reported in other nymphs or adults psyllids [20,27,31].

Response: Done. Thank you very much. I think the end of the sentence should be “…in other nymphal and adult psyllids [20,27,31].” Do you think so?

10. Lines 253-254: “The SCA has been found in many insects, including whiteflies [36] and psyllids [27].” Should be “The SCA has been observed in many insects, including whiteflies [36] and psyllids [27].”

Response: Done. Thank you very much.

11. Line 255: “The SCA was seemed to occur in…” should be “The SCA seems to occur in…”

Response: Done. Thank you very much.

12. Lines 259-260: “whereas the SCA with pores was involved in gustatory function were highly susceptible to humidity [52].” Should be “whereas the SCA with pores was involved in gustatory system as well as in humidity reception [52].”

Response: Done. Thank you very much.

13. Lines 270-271: “are related to the feeding behavior on the citrus with Las by these nymphs.” Should be “are related to feeding behavior of nymphs on Las-infected citrus.”

Response: Done. Thank you very much.

14. Lines 274-275: “psyllid's antennae and antennal sensilla and allow us to better understand the host location mechanism in D. citri.” Should be “psyllid's antennae and antennal sensilla and provides new insights on how D. citri nymphs might locate host plants.”

Response: Thank you very much for your suggestions. In general, the nymphs (can’t fly) may be not responsible for finding host plants, the adults locate host plants for feeding, oviposition and so on. “…allow us to better understand the host location mechanism in D. citri.”, for this sentence, we want to say that the development and changes of antennae and antennal sensilla in different stages of ACP (including nymphs and adults) may help us to learn about the host loction mechanism in D. citri from a developmental point of view. Maybe the expression is ambiguity, so we have changed the sentence as “…provides new insights on the olfactory behavior of D. citri nymphs.”

15. Table 2: “Abundance and distribution of sensilla on the antennae for the fifth nymphal stages of Diaphorina citri.” Should be “Abundance and distribution of sensilla on the antennae of the fifth nymphal stages of Diaphorina citri.”

Response: Done. Thank you very much.

16. Same for Table 3.

Response: Done. Thank you very much.

Special thanks to you for your kind suggestions.

Responds to Reviewer 3 (file “Revision 3”):

1. Page 2 (line 53): Coutinho-Abreu instead Coutnho-Abreu

Response: Done. Thank you very much.

2. Page 2 (line 55): Zanadi instead Zanardi

Response: Done. Thank you very much.

3. Page 3 (line 62): add “and” after [21]

Response: Done. Thank you very much.

4. Page 4 (line 92): means instead Means

Response: Done. Thank you very much.

5. Page 4 (line 105): I think that you refer to Fig 1D and 1E instead Fig 1C- please check

Response: Yes, thank you very much and we have done.

6. Page 5 (lines 129-138): you should provide information related to terminal hairs width. This information was presented in table, but not in text.

Response: Done. Thank you very much.

7. Page 5 (lines 134-135): the lengths of TH1 and TH2 in the second-instar nymphs were the shortest”: however, your mean comparison don’t say that: For TH1: only nymph 2 and nymph 5 are significantly different. For TH2: nymph5 is highe than nymphs 2 and 3. (please re-phrase the results based on the statistics, not based in numerical information). When writing width terminal hair, please do the same.

Response: Thank you very much for your kind comments. In this sentence, we want to point out that the shortest length of the TH1 and TH2, not for the statistics. Maybe our expressions are wrong, and we have rewritten the sentence according to your kind comments. You will find the information in page5 lines 133-141. Thank you very much for your kind comments and suggestions, it is very helpful and valuable for our MS and future researches.

8. Pages 5 and 6 (lines 141-145): you did not mention Fig 3B. Is this figure relevant? (You need to add in the text or delete the figure 3B).

Response: Done. Thank you very much.

9. Page 6 (line 150): Fig 4 (Are you referring to Fig 4A)? Because you did not mention this figure in results topic. Add Fig 4A in line 150 or delete Fig 4A.

Response: Yes, it should be Fig 4 A. Thank you very much.

10. Page 6 (line 153): Add Table 2 after “respectively”.

Response: Done. Thank you very much.

11. Page 18 (line 498): Table 1 caption: Add “and growth” after “Length”.

Response: Done. Thank you very much.

12. Page 20 (line 528): Table 3 caption: Terminal hairs (TH1 adn TH2) instead “TH1 and TH2”.

Response: Done. Thank you very much.

13. Fig 3: Cite Fig 3B in the text or delete it.

Response: Done. Thank you very much.

Special thanks to you for your comments and suggestions.

We tried our best to improve the manuscript and made some changes in the manuscript. We appreciate for Editors/Reviewers’ warm work earnestly, and hope that the correction will meet with approval.

Once again, thank you very much for your comments and suggestions.

---

## [Editor Report · Decision Letter 4]

19 May 2020

Morphological characterization of antennae and antennal sensilla of Diaphorina citri Kuwayama (Hemiptera: Liviidae) nymphs

PONE-D-19-23769R4

Dear Dr. Chen,

We are pleased to inform you that your manuscript has been judged scientifically suitable for publication and will be formally accepted for publication once it complies with all outstanding technical requirements.

With kind regards,

Walter S. Leal

Academic Editor

PLOS ONE
---

## [Editor Report · Acceptance letter]

21 May 2020

PONE-D-19-23769R4 

Morphological characterization of antennae and antennal sensilla of Diaphorina citri Kuwayama (Hemiptera: Liviidae) nymphs 

Dear Dr. Chen:

I am pleased to inform you that your manuscript has been deemed suitable for publication in PLOS ONE. Congratulations! Your manuscript is now with our production department. 

With kind regards,

on behalf of

Dr. Walter S. Leal 

Academic Editor

PLOS ONE